# ONLINE DECISION-FOCUSED LEARNING

**Aymeric Capitaine**[*]     **Maxime Haddouche**[†]     **Eric Moulines**[‡]

**Michael I. Jordan**[§]     **Etienne Boursier**[¶]     **Alain Durmus**[*]

## ABSTRACT

Decision-focused learning (DFL) is an increasingly popular paradigm for training predictive models whose outputs are used in decision-making tasks. Instead of merely optimizing for predictive accuracy, DFL trains models to directly minimize the loss associated with downstream decisions. However, existing studies focus solely on scenarios where a fixed batch of data is available and the objective function does not change over time. We instead investigate DFL in dynamic environments where the objective function and data distribution evolve over time. This setting is challenging for online learning because the objective function has zero or undefined gradients, which prevents the use of standard first-order optimization methods, and is generally non-convex. To address these difficulties, we (i) regularize the objective to make it differentiable and (ii) use perturbation techniques along with a near-optimal oracle to overcome non-convexity. Combining those techniques yields two original online algorithms tailored for DFL, for which we establish respectively static and dynamic regret bounds. These are the first provable guarantees for the online decision-focused problem. Finally, we showcase the effectiveness of our algorithms on a knapsack experiment, where they outperform two standard benchmarks.

## 1 INTRODUCTION.

Many real-world decision problems involve uncertainty, which is commonly handled through the predict-then-optimize framework (Bertsimas and Kallus, 2020). First, a prediction model is trained on historical data; then, its output is fed into an optimization problem to guide decision-making. This natural strategy has been successfully applied in many operation research (OR) problems, ranging from supply chain management (Acimovic and Graves, 2015; Fisher et al., 2016; Ban and Rudin, 2019; Bertsimas and Kallus, 2020), revenue management (Farias et al., 2013; Ferreira et al., 2016; Cohen et al., 2017; Chen et al., 2022) to healthcare operation (Bertsimas et al., 2013; Aswani et al., 2019; Gupta et al., 2020; Rath et al., 2017), see Mivsić and Perakis (2020) for an extensive review. It is clear that this approach would yield optimal decisions if the predictions were perfectly accurate. However, in practice, prediction errors are inevitable, and even small inaccuracies can propagate through the optimization process, potentially leading to poor decisions.

To address this limitation, an approach known as *decision-focused learning* (Mandi et al., 2024), also referred to as *smart predict-then-optimize* (Elmachtoub and Grigas, 2022) or *integrated learning-optimization* (Sadana et al., 2025), has emerged. Instead of optimizing for prediction accuracy alone, this method trains the predictive model to directly minimize the downstream decision loss. By aligning the learning objective with the decision-making goal, it produces models that are more robust to prediction errors in practical applications. While decision-focused learning yields strong empirical performance (Donti et al., 2017; Verma et al., 2022; 2023; Wang et al., 2023), theoretical development has so far been limited to the batch setting, where models are trained on pre-collected,

---

[*]Centre de Mathématiques Appliquées - CNRS, École Polytechnique, Paris, France
[†]INRIA - CNRS, Ecole Normale Supérieure, PSL Research University, France
[‡]EPITA, Laboratoire de Recherche de l'EPITA, Mohamed Bin Zayed University of AI, Paris, France
[§]Inria Paris, Ecole Normale Superieure PSL, UC Berkeley, Paris, France
[¶]Inria Saclay, Université Paris Saclay, LMO, Orsay, France

independently and identically distributed (i.i.d.) data (Wilder et al., 2019; Mandi et al., 2022; Shah et al., 2022; Schutte et al., 2024). This assumption breaks down in many real-world scenarios involving dynamic environments (Cheung et al., 2019; Padakandla et al., 2020) and shifting data distributions (Lu et al., 2018; Quiñonero-Candela et al., 2022).

Online learning (Cesa-Bianchi and Lugosi, 2006; Hazan, 2023) provides a general way to cope with such non-stationarity of data-generating processes. This framework considers a learner who makes decisions sequentially, at each round leveraging data collected from previous rounds to inform its next decision. Crucially, the objective function is allowed to vary over time, either in a stochastic or adversarial way. This provides a natural framework for the work that we present here, which extends decision-focused learning beyond the i.i.d., batch setting.

**Contributions.** We develop a theoretical foundation for online decision-focused learning, enabling its application in non-stationary settings. This presents a significant technical challenge, as the inherent difficulties of decision-focused learning, such as the non-differentiability of the objective function or the lack of convexity of the losses due to the bi-level nature of the problem, compound those already present in online learning. Our contributions are as follows:

(i) We formalize the online decision-focused learning problem by assuming that at each round, a decision-maker seeks to solve a linear optimization problem over a polytope but does not have access to the true cost function. Thus the decision-maker has to predict the cost using whatever partial information that it has at hand. The cost function is then revealed, and the decision-maker updates its model in a decision-focused fashion. This results in a bi-level optimization problem, where the inner problem consists in making a decision and the outer problem involves optimizing the resulting decision cost.

(ii) We present two algorithms to tackle this problem, *Decision-Focused Follow-the-Perturbed-Leader* (`DF-FTPL`) and *Decision-Focused Online Gradient Descent* (`DF-OGD`). While both rely on regularizing the inner problem to make the resulting decision differentiable, they differ on the way they update the parameters of the prediction model. `DF-FTPL` uses the FTPL approach (Hutter and Poland, 2005), and `DF-OGD` leverages a variant of Online Gradient Descent (Zinkevich, 2003). We establish sublinear convergence guarantees for both procedures, in the form of a static regret bound for the former and a dynamic regret bound for the latter. To our knowledge, these are the first provable guarantees for the online decision-focused learning problem.

(iii) Finally, we assess the performance of our algorithms on a knapsack experiment inspired by Mandi et al. (2024). Our simulations demonstrate that our approach outperforms the online version of two popular baselines, namely prediction-focused learning and Smart-Predict-then-Optimize, in both static and dynamic environments.

**Additional related work.** Several decision-focused approaches have been proposed in the batch setting, among which differentiating the associated KKT conditions Gould et al. (2016); Amos and Kolter (2017); Donti et al. (2017); Wilder et al. (2019); Mandi and Guns (2020), smoothing via random perturbation Berthet et al. (2020), building surrogate losses via duality Elmachtoub and Grigas (2022) or directional gradients Huang and Gupta (2024) and relying on pairwise ranking techniques Mandi et al. (2022), see Mandi et al. (2024); Sadana et al. (2025) for additional references. However, the extension of decision-focused learning to the online setting remains unexplored. While several recent studies address online bi-level optimization Shen et al. (2023); Tarzanagh et al. (2024); Lin et al. (2023), none of them are applicable to decision-focused learning as they rely on restrictive smoothness assumptions, that are incompatible with the structure of decision-focused problems. In particular, the decision-focused objective is typically non-convex and features gradients that are either zero or undefined, owing to its underlying linear structure.

This motivates the use of different online methods, specifically tailored to address these problems. On the one hand, lack of differentiability is usually tackled through zero-th order methods Héliou et al. (2020); Frezat et al. (2023), sub-gradient Duchi et al. (2011), proximal Dixit et al. (2019) or smoothing approaches Abernethy et al. (2014). On the other hand, methods to address non-convexity in online learning often rely on the existence of a near optimal oracle Kalai and Vempala (2005); Agarwal et al. (2019); Suggala and Netrapalli (2020); Xu and Zhang (2024) or additional smoothness conditions Lesage-Landry et al. (2020); Ghai et al. (2022). All those results are derived for *static regret* (Zinkevich, 2003) which compares the learned predictors to the best static strategy

and is defined in Section 2. A more challenging criterion, also described in Section 2 is *dynamic regret*, introduced in Zinkevich (2003) and later developed in Hall and Willett (2013); Besbes et al. (2015); Zhao et al.; Zhao and Zhang (2021) among others, which takes into account the evolution of the environment. An alternative approach to obtain guarantees in the online non-convex setting is to weaken static regret to *local regret* Hazan et al. (2017); Aydore et al. (2019); Zhuang et al. (2020); Hallak et al. (2021), which is the sum of the objective gradient norms evaluated in the iterates over time. Minimizing local regret corresponds to encouraging convergence toward stationary points. However as we shall see, this approach is not sensible in our framework, as objective gradients are either zero or undefined.

**Outline.** In Section 2, we introduce the online decision-focused problem, our notions of regret and our assumptions. In Section 3, we present the DF-FTPL and DF-OGD algorithms, before deriving bounds on the static regret of the former and the dynamic regret of the latter. Finally, we present our experiment in Section 4.

**Notation.** For a differentiable map $\varphi : \mathbb{R}^m \to \mathcal{Y}$ such that $\mathcal{Y} \subseteq \mathbb{R}$, $\nabla\varphi(x)$ denotes the gradient of $\varphi$ at $x \in \mathbb{R}^m$ and $\nabla^2\varphi(x)$ denotes the Hessian of $\varphi$. In the case where $\mathcal{Y} \subseteq \mathbb{R}^d$, $\nabla\varphi(x)$ denotes the Jacobian of $\varphi$. For two vectors $(v, w) \in \mathbb{R}^d \times \mathbb{R}^d$, $v \succcurlyeq w$ means that $v_i \geqslant w_i$ for any $i \in [d]$, and $\langle v, w \rangle = v^\intercal w$ refers to the standard Euclidian inner product. Also, $\|v\| = \sqrt{\langle v, v \rangle}$ is the standard euclidian norm. Given a compact convex set $\Theta \subseteq \mathbb{R}^d$, $\Pi_\Theta$ denotes the orthogonal projection onto $\Theta$. For a matrix $M \in \mathbb{R}^{m \times d}$, $\|M\|_{\mathrm{op}}$ refers to its L$^2$-operator norm. In the case $d = m$, $\lambda_{\min}(M)$ refers to its lowest real eigenvalue and $\lambda_{\max}(M)$ it largest real eigenvalue. For $(x, y) \in \mathbb{R}^2$, $x \propto y$ means that there exists $\lambda \in \mathbb{R}^\star$ such that $x = \lambda y$.

## 2 FRAMEWORK

**Sequential decision-making.** We consider an online decision-making problem over $T > 0$ periods, defined for any $t \in [T]$ as

$$\min_{w \in \mathcal{W}} \langle \bar{g}_t(X_t), w \rangle , \tag{1}$$

where $\mathcal{W} = \mathrm{Conv}(v_1, \ldots, v_K)$ is a bounded convex polytope of $\mathbb{R}^d$ with non-empty interior and vertices $\{v_i\}_{i=1}^K$. This feasible set appears naturally in many problems such as shortest-path (Gallo and Pallottino, 1988), portfolio selection (Li and Hoi, 2014) or mixed strategy design in games (Syrgkanis et al., 2015). In (1), $X_t$ are random covariates and $\bar{g}_t : \mathcal{X} \to \mathbb{R}^d$, is a deterministic cost function, which satisfies for instance $\bar{g}_t(X_t) = \mathbb{E}[Z_t \mid X_t]$ for some hidden state $Z_t \in \mathbb{R}^d$. At each period $t$, nature picks both a distribution for $X_t$ and a cost function $g_t$. This situation corresponds to the stochastic adversary setting (Rakhlin et al., 2011). Importantly, $X_t$ is revealed at the beginning of the round, but not $\bar{g}_t(X_t)$ which is only available at the end of the round. While $\bar{g}_t(X_t)$ is unknown to the decision-maker at the decision time, they have access to a family of models $g : \Theta \times \mathcal{X} \to \mathbb{R}^d$, parameterized by $\Theta \subset \mathbb{R}^m$, to predict it. The general form of the decision-making dynamics we consider can be described as follows: at each round $t \in [T]$, for a horizon $T \in \mathbb{N}$, given the current parameter $\theta_t$,

1. Nature picks a distribution for $X_t \in \mathcal{X}$ and a cost function $\bar{g}_t : \mathcal{X} \to \mathcal{Z}$.
2. The decision-maker observes $X_t$ and compute its prediction as $g(\theta_t, X_t)$.
3. Then, they take an action minimizing the resulting predicted cost

$$w_t = w_t^\star(\theta_t) \in \operatorname*{argmin}_{w \in \mathcal{W}} \langle g(\theta_t, X_t), w \rangle \tag{2}$$

4. Finally, the decision-maker observes $\bar{g}_t(X_t)$ and update $\theta_{t+1}$ for the next round.

Formally, the decision-maker considers a joint process $(\theta_t, w_t)_{t \in [T]}$, starting from $\theta_1 \in \Theta$ and a history $\mathcal{H}_0 = \emptyset$, and defined by the following recursion. At round $t \in [T]$, the decision-maker takes the best action given the current estimate $\theta_t$ and a new feature $X_t$ as in (2). Then, they observe $\bar{g}_t(X_t)$ and update their history $\mathcal{H}_t = \mathcal{H}_{t-1} \cup \{(X_t, \bar{g}_t(X_t), \theta_t, w_t^\star(\theta_t))\}$. Finally, they update their prediction parameter $\theta_{t+1}$ based on $\mathcal{H}_t$ through an algorithm $\mathsf{Alg}_t$. A central question for the decision-maker is then the choice of algorithms $\{\mathsf{Alg}_t\}_{t \in [T]}$ to fit their regression model, that is how to pick $\theta_t \in \Theta$ for each $t \in [T]$.

**Online decision-focused learning.** From an online perspective, *prediction-focused learning* consists in selecting at each $t \in [T]$ an algorithm $\mathsf{Alg}_t$ to estimate $\mathrm{argmin}_{\theta \in \Theta} \mathrm{R}_t(\theta)$, where $\mathrm{R}_t$ is a statistical risk based on the historical observations $\mathcal{H}_t$, typically chosen as the empirical risk $\mathrm{R}_t(\theta) = \sum_{i=1}^{t} \ell(\bar{g}_i(X_i), g(\theta, X_i))$, for some loss function $\ell$ (e.g., cross-entropy or squared error).

Alternatively, we consider in this paper, the *decision-focused learning* approach, which selects parameters $\theta \in \Theta$ by directly minimizing the downstream objective instead of a risk function. Specifically, the decision-maker chooses $\mathsf{Alg}_t$ to solve $\mathrm{argmin}_{\theta \in \Theta} \langle \bar{g}_t(X_t), w_t^\star(\theta) \rangle$ where $w_t^\star$ is defined in (2). From the previous formulation, we see that the decision-focused learning formulation corresponds to a bilevel optimization problem (Colson et al., 2007; Sinha et al., 2017; Ji et al., 2021).

We emphasize that no stationary assumptions are made on the process $\{X_t, \bar{g}_t(X_t)\}_{t \in [T]}$. In other words, the adversary may select any distribution for $X_t$ and any function $\bar{g}_t$ (as long as **H2** below is satisfied). To fully accommodate this flexibility, we assess the optimality of $\{\theta_t\}_{t \in [T]}$ with two notions of regret. Let

$$f_t : \theta \in \Theta \mapsto \langle \bar{g}_t(X_t), w_t^\star(\theta) \rangle , \tag{3}$$

denotes the loss incurred when taking an action based on the prediction parameter $\theta_t \in \Theta$. Following Zinkevich (2003), we consider the notions of static and dynamic regret to measure the effectiveness of a learning strategy $\{\theta_t\}_{t \in [T]}$, which are respectively:

$$\mathfrak{R}_T^s = \sum_{t \in [T]} f_t(\theta_t) - \inf_{\theta \in \Theta} \sum_{t \in [T]} \mathbb{E}[f_t(\theta)] \quad \text{and} \quad \mathfrak{R}_T^d = \sum_{t \in [T]} F_t(\theta_t) - \sum_{t \in [T]} \inf_{\theta \in \Theta} F_t(\theta), \tag{4}$$

where $F_t : \theta \mapsto \mathbb{E}[f_t(\theta) \mid \mathcal{H}_{t-1}]$. In our dynamic regret, the sequence of actions is compared against a sequence of oracles, each minimizing the instantaneous loss. Without taking the conditional expectation over $\mathcal{H}_{t-1}$, each comparator could overfit to the specific realization of $X_t$ resulting in an unrealistically strong benchmark. By considering the conditional expectation of the loss, we effectively regularize the dynamic comparators, making them meaningful competitors. Note that our static regret compare to the best fixed strategy, with respect to the averaged losses. This notion makes sense here as we aim to control those regrets in expectation over the randomness of the process.

We make the following mild regularity assumptions for the rest of the analysis.

**H1.** *(i)* $\Theta \subset \mathbb{R}^m$ *is a compact and convex set with diameter* $D_\Theta < \infty$. *(ii) for any* $t \in [T]$, $\theta \mapsto g(\theta, X_t) \in \mathbb{R}^{m \times d}$ *is continuously differentiable almost surely and* $\|\nabla_\theta g(\theta, X_t)\|_{\mathrm{op}} \leqslant G < \infty$ *for any* $\theta \in \Theta$ *almost-surely. (iii) For any* $t \in [T]$, $\|\bar{g}_t(X_t)\| \leqslant D_{\mathcal{Z}} < \infty$ *almost-surely.*

It is common in online learning to assume boundedness of the parameter space, model gradient and prediction space (Boyd, 2004; Bishop and Nasrabadi, 2006). Note that in the well-specified setting, *i.e.*, $\bar{g}_t = g(\bar{\theta}_t, \cdot)$ for some $\bar{\theta}_t$, these two former conditions automatically imply the latter. We emphasize however that we do not assume this realizability condition, in contrast to most of the literature on decision-focused learning (Bennouna et al., 2024).

Without further assumptions on the sequence of costs $\bar{g}_t$ and models $g$, the problem can still be made arbitrarily hard. Therefore, we make an assumption coming from the classification (Mammen and Tsybakov, 1999; Tsybakov, 2004) and bandit literature (Zeevi and Goldenshluger, 2009; Perchet and Rigollet, 2013) about the *margins* of the cost function. Recall that we denote by $\{v_i\}_{i=1}^{K}$ the vertices of $\mathcal{W}$. We define for any $x \in \mathcal{X}$ and $i \in [K]$, $u_i(\theta, x) = \langle g(\theta, x), v_i \rangle$.

**H2.** *There exist* $C_0 > 0$ *and* $\beta \in (0, 1]$, *such that almost surely for any* $t \in [T]$, $\theta \in \Theta$ *and* $\varepsilon \in [0, 1]$,

$$\mathbb{P}\left( \inf_{j \neq I_t(\theta)} \{u_j(\theta, X_t) - u_{I_t(\theta)}(\theta, X_t)\} \geqslant \varepsilon \mid \mathcal{H}_{t-1} \right) \geqslant 1 - C_0 \varepsilon^\beta ,$$

*where* $I_t(\theta) \in \mathrm{argmin}_{i \in [K]} u_i(\theta, X_t)$.

**H2** controls how difficult it is to solve problem (2) since it determines the objective gap between the optimal vertex $v_{I_t(\theta)} \in \mathcal{W}$ and the other vertices. In other words, it quantifies how identifiable is the optimal vertex, $\varepsilon = 0$ meaning it is not distinguishable from the others. If it is satisfied for $\beta \geqslant 1$, then it is automatically satisfied for $\beta = 1$ thus we do not loose any generality restraining $\beta$ to $[0, 1]$. This assumption is critical in our analysis, as it allows bounding the expected distance between the actual optimal decision $w_t^\star(\theta)$ and the regularized approximation $\tilde{w}_t(\theta)$ introduced in (5) below.

We expect **H**2 to hold in a wide variety of classical statistical settings. For example, in Appendix A we show that it is satisfied when $g : (\theta, X) \mapsto X\theta$ where $X$ has i.i.d standard Gaussian columns.

# 3   ONLINE ALGORITHMS FOR DECISION-FOCUSED LEARNING .

**On the need of regularization to get differentiation.**   From an online learning perspective, a natural approach to minimize the two regrets defined in (4) would be to update $\theta_{t+1}$ at each round $t$ through an algorithm $\mathsf{Alg}_t$ being either a variant of either the Follow-The-Leader algorithm (FTL, Kalai and Vempala, 2005) or Online Gradient Descent (OGD, Zinkevich, 2003), applied to the objective function $f_t$ defined in (3). However, these algorithms are designed for single-level optimization problems and require access to (sub-)gradients of the objective. In our setting, this requirement is problematic because the function $f_t$ does not admit an informative gradient: by construction, computing the gradient of $f_t$ involves differentiating the mapping $\theta \mapsto w_t^\star(\theta)$, which is generally not (sub-)differentiable. Indeed, $w_t^\star$ minimizes a linear function over the convex polytope $\mathcal{W}$, thus $\theta \mapsto w_t^\star(\theta)$ ranges in the finite set of vertices $\{v_1, \ldots, v_K\}$.

To address this issue, following Wilder et al. (2019), we propose to add a regularizer $\mathcal{R}$ to the objective function in (2). Accordingly, we define for any $\theta \in \Theta$ and some $\alpha_t > 0$

$$\tilde{w}_t(\theta) \in \operatorname*{argmin}_{w \in \mathcal{W}} \{\langle g(\theta, X_t), w \rangle + \alpha_t \mathcal{R}(w)\} , \tag{5}$$

which is a regularized approximation of $w_t^\star$. When this surrogate is continuously differentiable, which is the case for our choices of $\mathcal{R}$ in Section 3, we can use $\nabla \tilde{w}_t(\theta)$ in our algorithmic routine. This approach amounts to minimizing the surrogate

$$\tilde{f}_t : \theta \mapsto \langle \bar{g}_t(X_t), \tilde{w}_t(\theta) \rangle , \tag{6}$$

instead of $f_t$, which admits gradients of the form $\nabla \tilde{f}_t(\theta) = \nabla \tilde{w}_t(\theta)^\mathsf{T} \bar{g}_t(X_t)$. Note that there is a natural trade-off in the choice of the regularization parameter $\alpha_t$. On the one hand, choosing a large $\alpha_t$ makes the function $\tilde{w}_t(\theta)$ smoother. On the other hand, the larger $\alpha_t$ is, the more $\tilde{w}_t$ deviates from the true function $w_t^\star$ that we aim to approximate. As we will see later, it is possible to balance these two extremes by carefully tuning the parameter $\alpha_t$.

**On the choice of regularization on a general polytope.**   In what follows, we write $\mathcal{W}$ as the intersection of $n$ half-spaces, that is $\mathcal{W} = \{w \in \mathbb{R}^d : A^\mathsf{T} w - b \preccurlyeq 0\}$ where $n > 0$ is the number of faces, $A \in \mathbb{R}^{d \times n}$ and $b \in \mathbb{R}^n$. We assume that $\mathcal{W}$ is not degenerated, *i.e.* $AA^\mathsf{T} \in \mathbb{R}^{d \times d}$ is full-rank.

It remains to determine what regularizer $\mathcal{R}$ to choose in (6). We recall that our aim is to obtain a $\tilde{w}_t(\theta)$ in (5) which is differentiable for any $\theta \in \Theta$. A possible strategy is to choose $\mathcal{R}$ so $\tilde{w}_t$ remains in the strict interior of $\mathcal{W}$. In this case, $\tilde{w}_t(\theta)$ is differentiable in a neighborhood of any $\theta \in \Theta$ and admits a close-form Jacobian $\nabla \tilde{w}_t(\theta)$ by the implicit function theorem. A natural choice to force $\tilde{w}_t$ to remains in the interior of $\mathcal{W}$ is the corresponding log-barrier function

$$\mathcal{R} : w \mapsto - \sum_{i=1}^n \ln(b_i - A_i^\mathsf{T} w) , \tag{7}$$

where $A_i \in \mathbb{R}^d$ is the $i$-th column of $A$. With this choice of regularization, we show in Lemma 4 that $\nabla \tilde{w}_t$ has an explicit formulation, allowing its use in practice.

**Remark 1.** *We remark that in the special case where $\mathcal{W} = \{w \in \mathbb{R}^d : w \succcurlyeq 0 , \mathbf{1}^\mathsf{T} w = 1\}$ is the simplex of $\mathbb{R}^d$, an alternative choice for $\mathcal{R}$ in (5) is the negative entropy $\mathcal{R}_0 : w \mapsto \sum_{i \in [d]} w_i \ln(w_i)$. In this case, $\tilde{w}_t(\theta)$ in (5) reduces to the softmax mapping, which is differentiable and admits a close-form Jacobian. In Appendix C, we theoretically study the performances of our algorithms in this special case.*

**Approximate oracles to handle non-convexity.**   The bi-level structure of the problem yields unexpected properties. In particular, even when the regularizer $\mathcal{R}$ is strongly-convex in (26), which the case with both log-barriers and negative entropy, and the model $g$ is simple (such as $g(\theta, X) = X\theta$), $\tilde{f}_t : \theta \mapsto \langle \tilde{w}_t(\theta), \bar{g}_t(X_t) \rangle$ is lipschitz (see Lemmas 1 and 5) but not convex in general. This prevents us from directly using known online convex optimization algorithms (Hazan, 2023). However, a recent line of research (Agarwal et al., 2019; Suggala and Netrapalli, 2020; Xu and Zhang,

2024) has developed online algorithms in the non-convex case, provided the losses are Lipschitz-continuous. These studies combine near-optimal oracles with perturbation techniques to establish sublinear bounds on the expected regret. In line with this literature, we also assume to have access to an *approximate offline optimization oracle*, a notion that appeared in a slightly modified form in Suggala and Netrapalli (2020).

**Definition 1.** *An $\xi$-approximate offline optimization oracle (or approximate oracle), adapted to a class $\mathcal{C}$ is a mapping $\mathbf{O}_\xi$ taking a function $f \in \mathcal{C}$ such that $f : \Theta \to \mathbb{R}$, and outputting $\vartheta = \mathbf{O}_\xi(f) \in \Theta$ satisfying:*

$$f(\vartheta) \leqslant \inf_{\theta \in \Theta} f(\theta) + \xi . \tag{8}$$

This notion of an approximate oracle reflects the fact that, in non-convex settings, we cannot rely on subroutines that provably find a global minimizer of $\tilde{f}_t$ at time $t$ (as is possible in convex problems with offline gradient descent). Instead, we must be content with local minimizers. Their quality is measured by a parameter $\xi$, which becomes small in favorable loss landscapes. When $\mathcal{C}$ contains differentiable and Lipschitz functions, the stochastic gradient descent (SGD) algorithm can be used for the oracle $\mathbf{O}_\xi$. Indeed, a large body of work has shown that, even with models as complicated as deep neural networks, SGD can converge to local minimizers (Ghadimi and Lan, 2013; Mertikopoulos et al., 2020; Patel and Zhang, 2021; Cutkosky et al., 2023), justifying its use as an approximate oracle. A more detailed discussion is provided in Appendix B.

We conclude this remark by mentioning that it is possible to conduct a non-convex analysis without an approximate oracle by considering the weaker notion of *local regret* defined as $\sum_{t \in [T]} \|\nabla F_t(\theta_t)\|$; see (Hazan et al., 2017; Aydore et al., 2019; Zhuang et al., 2020; Hallak et al., 2021). However, such a definition is not meaningful in the DFL framework, where gradients are zero or undefined due to the structure of the problem.

The combined use of approximate oracles and regularization allows us to derive original online algorithms. In Section 3.1, we focus on a variant of the FTL algorithm, enjoying a static regret bound, while in Section 3.2 we propose a version of OGD enjoying a dynamic regret bound, both results holding in the non-convex case.

### 3.1 STATIC REGRET: DECISION-FOCUSED FOLLOW THE PERTURBED LEADER

Our first algorithm is displayed in Algorithm 1, where $\mathrm{Exp}(\eta)$ refers to an exponential distribution with parameter $\eta > 0$. It is inspired from the classic Follow-the-Leader approach (Kalai and Vempala, 2005), which consists in making a decision minimizing the sum of objective functions observed so far. When losses are non-convex, it is common to inject random noise at each step to regularize the total cost function, resulting in an approach known as Follow-the-Perturbed-Leader (Hutter and Poland, 2005; Suggala and Netrapalli, 2020). We employ this strategy as $\mathrm{Alg}_t$ to update $\theta_{t+1}$ in Algorithm 1. Note that the oracle in line 6 of Algorithm 1 crucially works with the *regularized* losses $\tilde{f}_1, \ldots, \tilde{f}_t$. This allows to use in practice gradient-based methods to obtain an approximate minimizer as per our discussion in Section 3, since these surrogate losses are differentiable and Lipschitz.

---

**Algorithm 1** Decision-Focused Follow The Perturbed Leader DF-FTPL

---

1: **Input:** horizon $T > 0$, initialization $\theta_1 \in \Theta$, $\xi$-approximate oracle $\mathbf{O}_\xi$ and history $\mathcal{H}_0 = \emptyset$.
2: **for** each $t \in \{1, \ldots, T\}$ **do**
3:     Observe $X_t \in \mathcal{X}$ and play $w_t^\star(\theta_t) = \mathrm{argmin}_{w \in \mathcal{W}} \langle g(\theta_t, X_t), w \rangle$ .
4:     Observe $\bar{g}_t(X_t) \in \mathcal{Z}$ and update the history $\mathcal{H}_t$.
5:     Draw $\sigma_t \in \mathbb{R}^d$ such that for all $j \in [d]$, the $j$-th component $\sigma_{j,t} \sim \mathrm{Exp}(\eta)$.
6:     Update

$$\theta_{t+1} = \mathbf{O}_\xi \left( \sum_{i=1}^t \tilde{f}_i - \langle \sigma_t, \cdot \rangle \right)$$

7: **end for**

---

Note that in line 3, the algorithm takes the best action $w_t^\star(\theta_t)$ given the predicted cost $g(\theta_t, X_t)$ (the regularized action $\tilde{w}_t$ is only needed to compute $\tilde{f}_t$). In most practical settings, $w_t^\star$ can be computed

efficiently with standard numerical solvers. However, we show in Appendix F that only being able to determine $\underline{w}_t(\theta_t) \in \mathcal{W}$ such that $\langle g(\theta_t, X_t), \underline{w}_t(\theta_t)\rangle - \langle g(\theta_t, X_t), w_t^\star(\theta_t)\rangle \leqslant \kappa$ for some $\kappa > 0$ only shifts the regret bounds of the next section by $\kappa$.

We provide a theoretical guarantee on the convergence of Algorithm 1 with the following static regret bound.

**Theorem 1.** *Assume $\boldsymbol{H}$ 1, $\boldsymbol{H}$ 2 and having access to an $\xi$-approximate oracle $\mathbf{O}_\xi$ adapted to $\{\sum_{i=1}^t \tilde{f}_i - \langle \sigma_t, \cdot\rangle\}_{t \in [T]}$. Let $\{\theta_t\}_{t \in [T]}$ be the output of DF-FTPL (Algorithm 1) instantiated with learning step $\eta > 0$ and regularization coefficients $\alpha_t = \alpha > 0$ for any $t$. Then:*

$$T^{-1}\mathbb{E}[\mathfrak{R}_T^s] = \tilde{\mathcal{O}}\left(\eta m^2 D \frac{1}{\alpha^2} + \frac{mD}{\eta T} + \xi + \alpha n\right),$$

*where $\mathbb{E}$ denotes the expectation on both data and the intrinsic randomness of DF-FTPL and $\tilde{\mathcal{O}}$ contains polynomial dependency in $\ln(1/\alpha), \ln(\ln(d))$.*
*Furthermore, taking $\eta \propto m^{1/4}T^{-3/4}n^{-1/2}$ and $\alpha \propto m^{3/4}n^{1/2}T^{-1/4}$ yields:*

$$T^{-1}\mathbb{E}[\mathfrak{R}_T^s] = \tilde{\mathcal{O}}\left(m^{3/4}n^{3/2}T^{-1/4} + \xi\right).$$

The proof of Theorem 1 can be found in Appendix G.2. In particular, Theorem 1 shows that DF-FTPL enjoys an average regret bound decaying in $T^{-1/4}$ as long as $\xi = \mathcal{O}(T^{-1/4})$. While it features a polynomial dependency $m^{3/4}$ on the dimension of $\Theta$, it only depends on the dimension of the decision space $\mathcal{W}$ through a $\ln\ln(d)$ term. This makes Algorithm 1 a particularly competitive approach when the decision space is of high dimension.

It is informative to compare our guarantee to existing bounds in the literature. Suggala and Netrapalli (2020), who also study non-convex online learning, achieves a rate of $\mathcal{O}(T^{-1/2})$ as long as their offline oracle satisfies $\xi = \mathcal{O}(T^{-1/2})$, at the cost of a degraded $m^{3/2}$ dependency on the dimension. Their faster rate in $T$ comes from the fact that they tackle a simpler, single-level problem as compared to our bi-level, non differentiable setting. Indeed, we need to regularize the inner problem to overcome non-differentiability as discussed in Section 3. This introduces an additional trade-off on the regularization strength $\alpha$ on top of the usual trade-off in the learning rate $\eta$, which is reflected in our rate.

### 3.2 DECISION-FOCUSED ONLINE GRADIENT DESCENT

While DF-FTPL (Algorithm 1) benefits from a converging static regret bound, the techniques involved in Suggala and Netrapalli (2020) are not enough to reach dynamic regret guarantees, which is particularly relevant in highly non-stationary environments where the optimal decisions may change significantly from one round to another. To this end, we go beyond the Follow-the-Leader approach and propose an original algorithm based on the celebrated Online Gradient Descent (Zinkevich, 2003). This procedure, which we call DF-OGD, is presented in Algorithm 2.

In Algorithm 2, at each time $t \in [T]$, a decision $w_t^\star(\theta_t) \in \mathcal{W}$ is made based on the current parameter $\theta_t \in \Theta$. The feedback $\bar{g}_t(X_t) \in \mathcal{Z}$ is then observed and used to define the surrogate objective $\tilde{f}_t$. Then, a near-minimizer of this objective is computed using the offline oracle from Definition 1. To address non-convexity, the gradient $\nabla \tilde{w}t(u_t)$ is evaluated at a point $u_t$ drawn uniformly at random from $[\theta_t, \vartheta_t]$. Finally, the parameter is updated to $\theta t + 1$ using the standard gradient step of OGD.

The main difference with Algorithm 1 is the update of $\theta_{t+1}$ through $\text{Alg}_t$. On the one hand, DF-FTPL invokes $\mathbf{O}_\xi$ to minimize the cumulative loss observed so far and perturbs the entire objective function. On the other hand, DF-OGD relies only on a near-optimizer of the most recent regularized cost, $\vartheta_t = \mathbf{O}_\xi(\tilde{f}_t)$, and perturbs the point at which the descent direction (gradient) is evaluated.

Moreover, note that Algorithm 1 is instanciated with a single pair of parameters $(\alpha, \eta)$ whereas Algorithm 2 uses on a sequence $(\alpha_t, \eta_t)_{t \in [T]}$. This additional flexibility is crucial for the algorithm to adapt to the variation of the problem so as to maintain a low dynamic regret.

We provide a convergence guarantee for Algorithm 2 with the following dynamic regret bound.

**Theorem 2.** *Assume $\boldsymbol{H}1$, $\boldsymbol{H}2$, access to a $\xi$-approximate oracle adapted to $\{\tilde{f}_t\}_{t \in [T]}$. Let $\{\theta_t\}_{t \in [T]}$ be the output of DF-OGD (Algorithm 2) instantiated with the non-increasing sequence $(\eta_t)_{t \in [T]}$ and*

---

**Algorithm 2** Decision-Focused Online Gradient Descent (`DF-OGD`)

---

1: **Input:** horizon $T > 0$, initialization $\theta_1 \in \Theta$ and history $\mathcal{H}_0 = \emptyset$.
2: **for** each $t \in \{1, \dots, T\}$ **do**
3:     Observe $X_t \in \mathcal{X}$ and play $w_t^\star(\theta_t) = \operatorname{argmin}_{w \in \mathcal{W}} \langle g(\theta_t, X_t), w \rangle$ .
4:     Observe $\bar{g}_t(X_t) \in \mathcal{Z}$ and update the history $\mathcal{H}_t$.
5:     Get $\vartheta_t = \mathbf{O}_\xi(\tilde{f}_t)$.
6:     Draw $\delta_t \sim \operatorname{Unif}([0, 1])$, compute $u_t = \vartheta_t + \delta_t(\theta_t - \vartheta_t)$ and $\tilde{\nabla}_t(u_t) = \nabla \tilde{w}_t(u_t)^\intercal \bar{g}_t(X_t)$ .
7:     Update $\theta_{t+1} = \Pi_\Theta(\theta_t - \eta_t \tilde{\nabla}_t(u_t))$ .
8: **end for**

---

*regularization coefficients* $(\alpha_t)_{t \in [T]}$. *Then:*

$$T^{-1}\mathbb{E}\big[\mathfrak{R}_T^d\big] = \tilde{\mathcal{O}}\left(\mathbb{E}\left[\frac{1 + P_T}{T\eta_T} + \frac{1}{T}\sum_{t \in [T]}\frac{\eta_t}{\alpha_t^2} + n\alpha_t\right] + \xi\right) .$$

*where* $P_T = \sum_{t=1}^{T-1}\|\vartheta_{t+1} - \vartheta_t\|$, $\mathbb{E}$ *denotes the expectation on both data and the intrinsic randomness of* `DF-OGD` *and* $\tilde{\mathcal{O}}$ *contains polynomial dependency in* $\ln(1/\alpha), \ln(\ln(d))$.

*Furthermore, assume* $(t^{-1}(1 + P_t))_{t \geqslant 1}$ *is non-increasing almost surely with* $P_t = \sum_{s=1}^{t-1}\|\vartheta_{s+1} - \vartheta_s\|$. *Then, using* $\alpha_t \propto n^{-1/2}t^{-1/4}(1 + P_t)^{1/4}$ *and* $\eta_t \propto n^{-1/2}t^{-3/4}(1 + P_t)^{3/4}$ *for any* $t \in [T]$ *leads to:*

$$T^{-1}\mathbb{E}\big[\mathfrak{R}_T^d\big] = \tilde{\mathcal{O}}\Big(\mathbb{E}\Big[\sqrt{n}(1 + P_T)^{1/4}T^{-1/4}\Big] + \xi\Big) .$$

Our dynamic regret bound naturally depends on $P_T$, which captures the problem's variability by measuring the cumulative distance between approximate minimizers over time. When $\xi = \mathcal{O}((1 + P_T)^{1/4}T^{-1/4})$, the average dynamic regret decreases at the rate $\mathcal{O}((1 + P_T)^{1/4}T^{-1/4})$. Notably, the bound is independent of the dimension of $\Theta$, and depends only mildly on the dimension of $\mathcal{W}$ through a $\ln\ln(d)$ factor. This constitutes a key strength of the optimistic strategy underlying `DF-OGD`, making it particularly well-suited for high-dimensional settings.

It is instructive to compare our guarantee with those established in recent studies on dynamic regret. For instance, Zhang et al. (2018) derive a $\mathcal{O}(T^{-1/2}\sqrt{T(P_T + 1)})$ bound. However, their setting is not directly comparable to ours, as they consider a simpler single-level problem with differentiable and convex objectives. In contrast, our framework involves non-convex, non-differentiable losses due to the bi-level nature of decision-focused learning. More recently, Huang and Wang (2025) obtained a $\mathcal{O}((1 + P_T^\infty)^{1/3}T^{-1/3})$ bound, where $P_T^\infty := \sum_{t=1}^T \|f_{t+1} - f_t\|_\infty$. Yet, this rate is achieved under substantially more favorable conditions: a single-level problem with losses that are strongly convex or Lipschitz, an additional assumption of "quasi-stationary" and a more challenging path involving the full landscapes of the $f_t$s. By comparison, our losses are neither Lipschitz (indeed, they may even be discontinuous due to the linearity of the lower-level problem) nor required to be stationary over time—the only restriction being **H**2 to hold. We view the ability of our algorithm to achieve efficient convergence despite these demanding conditions as a core contribution of our work.

## 4 EXPERIMENTS

In this section, we compare the performances of our algorithms `DF-FTPL` and `DF-OGD` to two important benchmarks, namely prediction-focused learning and SPO (Elmachtoub and Grigas, 2022).

**Setting.** Our experimental setup is inspired by the knapsack example from Mandi et al. (2024). More precisely, we consider a decision maker who must pick at each $t \in [T]$ an object $v_t \in \mathcal{V}$ among $K$ items denoted $\mathcal{V} = \{1, 2, \dots, K\}$ with respective costs $\bar{g}_t(X) = (\bar{g}_{t,1}(X), \dots, \bar{g}_{t,K}(X)) \in [0, 1]^K$ depending on some covariates $X \in \mathcal{X}$. At the beginning of each period, the decision-maker only observe covariates $X_t \in \mathcal{X} \subset \mathbb{R}^p$, and have at their disposal a parametric model $g$ :

$\Theta \times \mathcal{X} \to [0,1]^K$ to predict $\bar{g}_t$. Given their current parameter $\theta_t \in \Theta$, they predict item costs as $g(\theta_t, X_t) = (g_1(\theta_t, X_t), \ldots, g_K(\theta_t, X_t))$ and pick an item $v_t^\star(\theta_t) = \operatorname{argmin}_{i \in [K]} g_i(\theta_t, X_t)$. This setting is depicted in Figure 2 for $K = 2$ in Appendix E. After having made their decision, the decision-maker observes the true item costs $\bar{g}_t(X_t)$, and update $\theta_{t+1}$ for the next round based on this feedback. Note that this setting can directly be mapped in the simplex example of Remark 1, since $v_t^\star(\theta_t) = \operatorname{argmin}_{w \in \mathcal{W}_0} \langle w, g(\theta_t, X_t) \rangle$, $\mathcal{W}_0$ being the simplex of $\mathbb{R}^K$. For this reason, we instantiate our algorithms DF-OGD and DF-FTPL with the negative entropy regularizer, following Remark 1.

**Synthetic data.** We instantiate the previous problem with the following synthetic data. For any $t \in [T]$, we draw $X_t \in \mathbb{R}^{K \times p}$ with correlated rows, and generate a cost vector $c_t(X_t) \in \mathbb{R}^k$ as :

$$c_t(X_t) = A \sin^4((2X_t \theta_t^\star)^{-1}) + \varepsilon_t , \tag{9}$$

where $A > 0$, $\varepsilon_t \sim \mathrm{N}(0, I_K)$ is a Gaussian noise and $\theta_t^\star \in \mathbb{R}^p$ is a parameter which satisfies $\theta_t^\star = 1/2\,\theta^\star + 1/2\,\zeta_t$, where $\zeta_t \sim \mathrm{N}(0, I_p)$ for some $\theta^\star \in \mathbb{R}^p$. This is a challenging data generating process, since $\bar{g}_t(X_t) = \sin^4((2X_t^\top \theta_t^\star)^{-1})$ is non-stationary and highly non-linear, and features are correlated. Equation (9) is discussed more in detail in Appendix E. To predict $c_t$ from $X_t$, we assume that the decision-maker has access to a class of linear predictors of the form $g : (\theta, X) \mapsto X\theta$.

**Benchmarks.** In this setting, we compare the performances of DF-FTPL (Algorithm 1) and DF-OGD (Algorithm 2) to two benchmarks. First, we implement Prediction-Focused Online Gradient Descent (PF-OGD). This strategy consists in training in an online manner the model $g$ at each timestep so it minimizes the statistical loss $\ell^{\mathrm{mse}} : (v, X\theta) \mapsto \|v - X\theta\|^2$, irrespective of the downstream decision problem. Then, decisions are greedily made based on the predictions of the model. This approach is formally described in Algorithm 3 in Appendix E. Second, we compare our algorithms to an online version of the Smart Predict-then-Optimize (online SPO, see Algorithm 4) approach from Elmachtoub and Grigas (2022). This algorithm introduces a differentiable and convex loss which upper-bounds the actual decision-focused loss. Given its effectiveness and versatility, it is considered as a very important benchmark in the literature.

**Results.** In Figure 1, we plot on the left-hand panel the average cumulated cost $t \mapsto t^{-1} \sum_{s=1}^t \bar{g}_{s, v_s}(X_s)$ incurred by DF-OGD, DF-FTPL, PF-OGD and online SPO over 10 runs of $T = 5000$ timesteps, as well as the associated 95% confidence intervals. On the right-hand panel, we plot the average Mean Square Error (MSE) resulting from the sequence of prediction parameters $(\theta_t)_{t \in [T]}$. It appears that both DF-FTPL and DF-OGD outperform PF-OGD and online SPO from a decision point of view, while incurring a higher MSE. This is in line with decision-focused approach, which cares about decision cost rather than statistical loss. This experiment shows that *(i)* DFL outperforms PFL when models are clearly mispecified and *(ii)* our algorithms also outperforms the celebrated online SPO. We also mention the presence of additional numerical experiments in

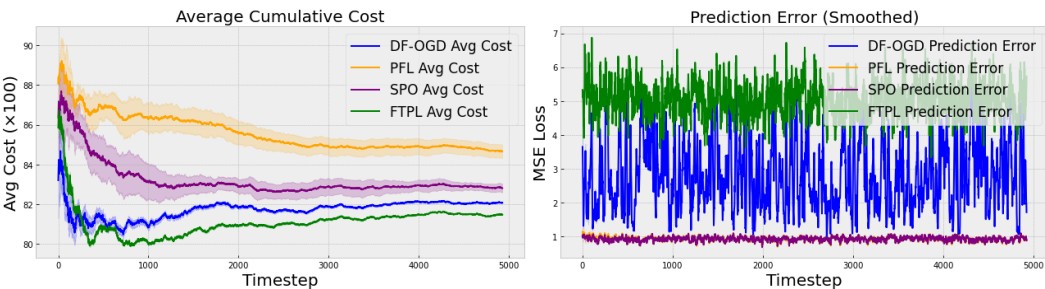

Figure 1: Average cumulated cost (left) and prediction errors (right) of DF-OGD, PF-OGD, DF-FTPL and online SPO.

Appendix E, to study how model misspecification affects the relative performances of DFL and PFL.

## 5 CONCLUSION.

Decision-focused learning offers a promising way to integrate prediction into decision making. We extend its analysis from the batch to the online setting, enabling non-stationary data and varying objectives. Our algorithms DF-FTPL and DF-OGD comes with provable guarantees, and empirically outperform prediction-focused learning and online SPO in our experiment.

We believe that this work can be extended in several ways. First, it is plausible that faster rates (e.g., $T^{-1/2}$) could be obtained using more traditional online-learning techniques that bypass the bilevel framework, such as discretizing the parameter space and applying expert aggregation over the resulting bins. However, such approaches would likely incur a prohibitive dependence on the parameter dimension $m$ (e.g., exponential in $m$). Exploring this direction remains however an interesting avenue in future work. Second, we believe that alternative smoothing techniques could be successfully employed, such as Moreau-Yosida transform and proximal operators, to adress the more challenging of a general convex set $\mathcal{W}$. Third, it would be valuable to investigate less adversarial environments, such as those involving i.i.d. data, to explore whether stronger theoretical guarantees could be obtained and whether novel algorithmic designs might emerge. Finally, implementing our method in more ambitious experimental settings would be of great interest from a practical perspective.

## ACKNOWLEDGMENT

Funded by the European Union (ERC, Ocean, 101071601). M.H. is partially supported by the European Research Council Starting Grant DYNASTY – 101039676. Views and opinions expressed are however those of the author(s) only and do not necessarily reflect those of the European Union or the European Research Council Executive Agency. Neither the European Union nor the granting authority can be held responsible for them.

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

## A   H2 IS SATISFIED IN THE GAUSSIAN LINEAR CASE.

In this section, we prove that **H**2 holds true when $\mathcal{W}$ is the simplex of $\mathbb{R}^d$, $g$ is a linear model and $X$ has columns distributed according to a Gaussian distribution. In what follows, we denote by $X_j \in \mathbb{R}^d$ the $j$-th column of $X$.

**Proposition 1.** *Assume that $g : (\theta, X) \in \Theta \times \mathcal{X} \mapsto X\theta$ where $\Theta = \{\theta \in \mathbb{R}^m : \|\theta\|_2 = 1\}$ and $X = (X_1 \mid \ldots \mid X_m) \in \mathbb{R}^{d \times m}$ where $X_j \overset{i.i.d.}{\sim} \mathcal{N}(\mathbf{0}, \boldsymbol{I})$. Then, **H**2 is true for any $\theta \in \Theta$ with*

$$C_0 = \frac{d(d-1)}{2\sqrt{\pi}} \quad and \quad \beta = 1 .$$

*Proof.* Let $\theta \in \Theta$. In what follows, we denote $i = I(\theta)$ where $I(\theta)$ is defined as in **H**2. Since $\mathcal{W}$ is the simplex of $\mathbb{R}^d$, $v_j$ is the $j$-th vector of the canonical basis of $\mathbb{R}^d$. For any $\varepsilon > 0$ we have:

$$\mathbb{P}\left(\inf_{j \neq i} |\langle X\theta, v_j \rangle - \langle X\theta, v_i \rangle| \leqslant \varepsilon\right) \leqslant \mathbb{P}\left(\bigcup_{j \neq i} \{|\langle X\theta, v_j \rangle - \langle X\theta, v_i \rangle| \leqslant \varepsilon\}\right)$$

$$\leqslant \sum_{j \neq i} \mathbb{P}(|\langle \theta, X^{\mathsf{T}}(v_j - v_i) \rangle| \leqslant \varepsilon)$$

$$= \sum_{j \neq i} \mathbb{P}(|\langle \theta, X_j - X_i \rangle| \leqslant \varepsilon) . \tag{10}$$

Since for any $j \neq i$, $\langle \theta, X_j - X_i \rangle \sim \mathcal{N}(0, 2\|\theta\|_2^2)$ with density $f_j$, we have:

$$\mathbb{P}(|\langle \theta, X_j - X_i \rangle| \leqslant \varepsilon) = \int_{-\varepsilon}^{\varepsilon} f_j(x) \, \mathrm{d}x = \left(2\sqrt{\pi}\|\theta\|_2\right)^{-1} \int_{-\varepsilon}^{\varepsilon} \exp\left(-\frac{x^2}{4\|\theta\|_2^2}\right) \mathrm{d}x$$

and since the integrand reaches its maximum in $x = 0$,

$$\leqslant \varepsilon\left(\sqrt{\pi}\|\theta\|_2\right)^{-1} . \tag{11}$$

Then, plugging (11) in (10) and summing over all pairs $(i, j)$ such that $j \neq i$ yields the desired result. $\qquad\square$

Note that the $d^2$ factor in $C_0$ comes from the use of an union bound, and could be reduced with a refined analysis.

## B   EXTENDED DISCUSSION ON THE RELEVANCE OF APPROXIMATE ORACLES.

The notion of an approximate oracle reflects the fact that, in non-convex settings, we cannot rely on subroutines that provably reach a global minimizer of $\tilde{f}_t$ at time $t$ (as offline gradient descent would in convex problems). Instead, we must settle for local minimizers, whose quality is governed by a parameter $\xi$—which vanishes in favorable loss landscapes.

As a concrete example, consider **O** as the stochastic gradient descent (SGD) algorithm. Even in the context of deep networks, it is plausible that **O** converges to a local minimizer. Indeed, a large body of work has analyzed SGD in non-convex settings, establishing convergence to stationary points either in expectation (Ghadimi and Lan, 2013) or almost surely (Mertikopoulos et al., 2020; Patel and Zhang, 2021; Cutkosky et al., 2023). Such stationary points may correspond to local/global minima or saddle points.

Recent studies further show that SGD avoids saddle points. For instance, Mertikopoulos et al. (2020) proved that the trajectories of SGD almost surely avoid all strict saddle manifolds—i.e., sets of critical points $x$ where the Hessian has at least one negative eigenvalue. These manifolds include connected families of non-isolated saddle points, a phenomenon common in the loss landscapes of overparametrized neural networks (Li et al., 2018).

Beyond SGD, similar guarantees extend to more general methods. In particular, the stochastic Riemannian Robbins–Monro method (a broad template encompassing various algorithms) has been shown to converge almost surely to local or global minima (Hsieh et al., 2023).

Finally, we highlight that, in most practical industrial settings, solving the downstream optimization problem is typically far more computationally demanding than updating the prediction model. It is therefore reasonable to assume that the oracle call represents only a small fraction of the overall computational cost.

## C   Supplementary results on the simplex.

We now provide convergence guarantees for Algorithms 1 and 2 when $\mathcal{W}$ is the simplex. To do so we choose another regularizer than done in the main document.

### C.1   Additional framework

We focus on the case where $\mathcal{W}$ is the simplex of $\mathbb{R}^d$, that is

$$\mathcal{W}_0 = \left\{ w \in \mathbb{R}^d : w \succcurlyeq 0, \ \mathbf{1}^\intercal w = 1 \right\} .$$

This setting encompasses various decision-making scenarios such as portfolio selection Li and Hoi (2014) or mixed strategy design in multiplayer games Syrgkanis et al. (2015). In this case, we choose the negative entropy

$$\mathcal{R}_0 : w \mapsto \sum_{i \in [d]} w_i \ln(w_i) ,$$

as the regularizer in (5). The minimizer $\tilde{w}_t(\theta) \in \mathcal{W}_0$ in (5) with $\mathcal{R} = \mathcal{R}_0$ can easily be shown to be the softmax mapping, that is it satisfies for any $i \in [d]$:

$$\tilde{w}_{t,i}(\theta) = \frac{\exp(-\alpha^{-1} g_i(\theta, X_t))}{\sum_{k \in [d]} \exp(-\alpha^{-1} g_k(\theta, X_t))} .$$

It is clear from this expression that $\tilde{w}_t$ is differentiable, and that for any $\theta \in \Theta$:

$$\nabla \tilde{w}_t(\theta) = -\frac{1}{\alpha} [\text{diag}[\tilde{w}_t(\theta)] - \tilde{w}_t(\theta)\tilde{w}_t(\theta)^\intercal] \nabla_\theta g(\theta, X_t) . \tag{12}$$

### C.2   Results

**DF-FTPL**

**Proposition 2.** *Assume H1, H2 and having access to an $\xi$-approximate optimization oracle $\mathbf{O}_\xi$ adapted to $\left\{ \sum_{i=1}^t \tilde{f}_i - \langle \sigma_t, \cdot \rangle \right\}_{t \in [T]}$. Fix $\mathcal{W} = \mathcal{W}_0$, $\mathcal{R} = \mathcal{R}_0$. Let $\{\theta_t\}_{t \in [T]}$ be the output of* DF-FTPL *(Algorithm 1) instantiated with learning step $\eta > 0$ and regularization coefficients $\alpha_t = \alpha > 0$ for any $t$. Then:*

$$T^{-1}\mathbb{E}[\mathfrak{R}_T^s] \leqslant \tilde{\mathcal{O}}\left( \eta m^2 D \frac{1}{\alpha^2} + \frac{mD}{\eta T} + \xi + \alpha \ln(d) \right) ,$$

*where $\mathbb{E}$ denotes the expectation on both data and the intrinsic randomness of* DF-FTPL *and $\tilde{\mathcal{O}}$ contains polynomial dependency in $\ln(1/\alpha), \ln(\ln(d))$.*

The proof of this result is postponed to Appendix G.4.

**DF-OGD**

**Proposition 3.** *Assume H1, H2, and having access to an $\xi$-approximate optimization oracle $\mathbf{O}_\xi$ adapted to $\left\{ \tilde{f}_i \right\}_{t \in [T]}$. Fix $\mathcal{W} = \mathcal{W}_0$, $\mathcal{R} = \mathcal{R}_0$. Let $\{\theta_t\}_{t \in [T]}$ be the output of* DF-OGD *(Algorithm 2)*

*instantiated with the non-increasing sequence $(\eta_t)_{t\in[T]}$ and regularization coefficients $(\alpha_t)_{t\in[T]}$. Then:*

$$T^{-1}\mathbb{E}\big[\mathfrak{R}_T^d\big] \leqslant \frac{D_\Theta^2 + D_\Theta\mathbb{E}[P_T]}{2T\eta_T} + \sum_{t\in[T]}\frac{25D_{\mathcal{Z}}G\eta_t}{32T\alpha_t^2}$$

$$+ 2D_{\mathcal{Z}}\sum_{t\in[T]}\alpha_t[1 + 2\ln(d)C_0\{\ln(2\alpha_t^{-1}) + (1-\beta)\ln^2(\alpha_t\ln d)\}] + \xi \,,$$

*where $P_T = \sum_{t=1}^{T-1}\|\vartheta_{t+1} - \vartheta_t\|$.*

The proof of this result is postponed to Appendix G.5.

Note that the principal gain, compared to the general polytope case, is to avoid the dependency in $n$ the number of faces and enjoys a nice $\ln(d)$ dependency in the dimension of the decision space.

## D    BACKGROUND ON THE FTPL ALGORITHM

**Approximate Optimization Oracle.**    The recent work of Suggala and Netrapalli (2020) proposed an online algorithm for nonconvex losses $(\ell_t)_{t\in[T]}$ with static regret guarantees. They rely on an *approximate optimization oracle* $\mathcal{O}$ which takes as input any function $\ell$, a $d$-dimensional vector $\sigma$ and returns an approximate minimizer of $\ell - (\sigma, \cdot)$. An optimization oracle is called "( $\xi, \chi$ )-approximate optimization oracle" if it returns $\mu^* \in \mathcal{K}$ such that

$$\ell(\mu^*) - \langle\sigma, \mu^*\rangle \leqslant \inf_{\mu\in\mathcal{K}}[\ell(\mu) - \langle\sigma, \mu\rangle] + (\xi + \chi\|\sigma\|_1) \,. \tag{13}$$

We denote the output of such an optimization oracle by $\mu^* = \mathcal{O}_{\xi,\chi}(\ell - \langle\sigma, \cdot\rangle)$. Note the notion of oracle described in (13) is very close from the Definition 1 we made in Section 3 on the expert sequence $(\vartheta_t)_{t\in[T]}$ (in this case $\chi = 0$ as no $\sigma$ is involved).

**Remark 2.** *Note that, in our work, we made the choice to fix $\chi = 0$. This stronger assumption is due to the will of having a unifying framework encompassing both the setup for* `DF-FTPL` *and* `DF-OGD`*.*

**Follow The Perturbed Leader.**    Given access to an $(\xi, \chi)$-approximate optimization oracle, Suggala and Netrapalli (2020) study the FTPL algorithm which is described by the following recursion. Starting from $\hat{\mu}_1$, at each time steps $t$, $\hat{\mu}_t \in \mathcal{K}$ is updated as follows:

$$\hat{\mu}_{t+1} = \mathcal{O}_{\xi,\chi}\left(\sum_{i=1}^{t-1}\ell_i - \langle\sigma_t, \cdot\rangle\right) \tag{14}$$

where $\sigma_t \in \mathbb{R}^d$ is a random perturbation such that $\sigma_{t,j}$, the j-th coordinate of $\sigma_t$, is sampled from $\mathrm{Exp}(\eta)$, the exponential distribution with parameter $\eta$.

Following this update route, the addition of an exponential noise allowing to handle the nonconvexity of the losses, they reach the following static regret bound.

**Proposition 4** (Theorem 1 of Suggala and Netrapalli (2020))**.** *Let $D$ be the diameter of $\mathcal{K}$. Suppose that $\ell_t$ is L-Lipschitz w.r.t L1 norm, for all $t \in [T]$. For any fixed $\eta$, FTPL (Equation (14)) with access to a $(\xi, \chi)$-approximate" optimization oracle satisfies the following static regret bound:*

$$\frac{1}{T}\left(\sum_{t=1}^{T}\ell_t(\hat{\mu}_t) - \inf_\mu\sum_{t=1}^{T}\ell_t(\mu)\right) \leqslant \mathcal{O}\left(\eta m^2 DL^2 + \frac{m(\chi T + D)}{\eta T} + \xi + \chi mL\right),$$

*where $\mathbb{E}$ denotes the expectation over $\sigma_1\cdots\sigma_T$.*

## E    ADDITIONAL EXPERIMENTAL MATERIAL

**The motivating example from Mandi et al. (2024).**    In this paragraph, we recall the example that motivates our experiment. Mandi et al. (2024) illustrates the interest of decision-focused learning

with a simple problem where a decision maker seeks to pick, between two objects, the one with the lowest cost. Before picking an object, they do not know costs but have at their disposal a model to predict it. This situation is depicted in Figure 2. For instance, if the true cost $\bar{g}_t = (\bar{g}_{t,1}, \bar{g}_{t,K})$ is (∗), any prediction $\hat{g}_t = (\hat{g}_{t,1}, \hat{g}_{t,K})$ lying in the blue-shaded area, such as (+), induces the optimal decision $v_t = 1$. As observed in Mandi et al. (2024), a prediction-focused approach may underperform compared to a decision-focused one in this setting. This is because generating a prediction $\hat{v}_t$ that closely approximates the true value $v_t$—in the sense of minimizing statistical loss—does not necessarily ensure that $\hat{v}_t$ falls on the same side of the 45° line as $v_t$. For example, the prediction (×) is satisfactory from a predictive point of view, but induces a sub-optimal action. On the contrary, one would expect the decision-focused approach to produce predictions that lie further away from (∗) since it does not minimize prediction error, but on the right side of the 45° line; see for instance (▲) or (▼) on Figure 2.

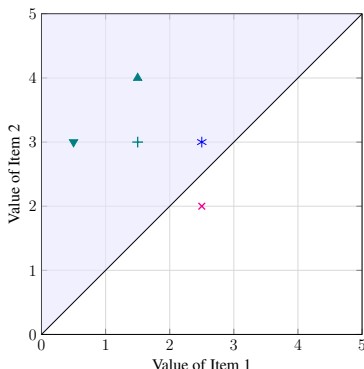

Figure 2: Figure 2 from Mandi et al. (2024).

**Details on the experimental setup.** In this paragraph, we provide more detail about the experimental setup in Section 4 and perform additional numerical simulations. We start by explaining more precisely how the data used in our experiment are drawn. First, for any $t \in [T]$, $X_t \in \mathbb{R}^{K \times p}$ is constructed as follows: (i) we generate a Toeplitz covariance matrix $\Sigma = (\rho^{|i-j|})_{(i,j) \in [K]^2}$ for some $\rho \in (0, 1)$, (ii) apply a Cholesky decomposition to it: $\Sigma = LL^\top$, and (iii) generate a matrix $\bar{X}_t$ with standard Gaussian entries. Then, $X_t$ is defined as $X_t = L\bar{X}_t$. This introduces correlation between features, which makes convergence harder for an online linear model. Second, we generate $v_t \in \mathbb{R}^K$ as follows:

$$c_t = \min(\max(\tilde{c}_t, 0), 1) \quad \text{with} \quad \tilde{c}_t = A \sin^4((2X_t \theta_t^\star)^{-1}) + \varepsilon .$$

In the above equation, $A > 0$ is a constant, $\theta_t^\star$ satisfies:

$$\theta_t^\star = \frac{1}{2}\theta^\star + \frac{1}{2}\zeta_t , \quad \text{where} \quad \zeta_t \sim \mathrm{N}(0, I_K) \quad \text{and} \quad \theta^\star \in \mathbb{R}^K ,$$

and $\varepsilon_t \sim \mathrm{N}(0, I_K)$ is Gaussian noise. The fact that $\theta_t^\star$ varies throughout time and that the relationship between $v_t$ and $X_t$ is highly linear makes learning hard for a linear model.

We now present the values used for the different parameters in our experiment. We consider $K = 5$ items and $p = 10$ features. The horizon is set to $T = 5000$. For each plot, we run $N = 10$ times DF-OGD and PF-OGD. The reported error bars are 95% Gaussian confidence intervals (from the CLT). In (9), $A = 45$ and the correlation coefficient for $X_t$ is $\rho = 0.8$. Our algorithms are instantiated with the following parameters. First, DF-OGD runs with $(\alpha_t)$ and $(\eta_t)$ as suggested by our theoretical analysis. The oracle from **H**2 is obtained through a SGD algorithm performing $t_\vartheta = 10$ steps at each iteration with a learning rate set to $\eta_\vartheta = 0.01$, while being initialized at $\vartheta_{t-1}$. On the other hand, PF-OGD runs with a learning rate $\eta = 10$ determined by grid search. Our code is implemented with pytorch.

**Benchmark algorithms.**

---

**Algorithm 3** Prediction-Focused Online Gradient Descent (`PF-OGD`).

---

1: **Input:** horizon $T > 0$, initialization $\theta_1 \in \Theta$.
2: **for** each $t \in \{1, \ldots, T\}$ **do**
3:     Observe $X_t$, predict $c_t = g(\theta_t, X_t)$
4:     Play $v_t^\star(\theta_t) = \operatorname{argmin}_{i \in [K]} c_{t,i}$ , observe $\bar{g}_t(X_t)$.
5:     Update $\theta_{t+1} = \Pi_\Theta(\theta_t - \eta \nabla \ell^{\mathrm{mse}}(\bar{g}_t(X_t), X_t \theta_t))$ .
6: **end for**

---

**Deviation from realizability.** As argued in Section 1, prediction-focused learning would be optimal if the prediction model made no mistake, since in this case problem (2) would perfectly align with (1). As a consequence, one would expect prediction-focused learning to perform very well in the realizable setting, that is when for any $t \in [T]$, there exists $\bar{\theta}_t \in \Theta$ such that $\bar{g}_t(X_t) = g(\bar{\theta}_t, X_t)$. On the flip side, it is likely to struggle in a mispecified setting. This is in contrast with `DF-OGD`, whose theoretical performance analysis does not highlight any peculiar dependence on realizability. We therefore hypothetize that PFL should outperform DFL in the realizable case, and DFL should take the upper hand when the prediction model is highly mispecified. To test this hypothesis, we run an alternative experiment where we interpolate between the well-specified, linear case, and ill-specified, non-linear one. More precisely, for $\gamma \in [0, 1]$, we generate $(X_t, v_t^{(\gamma)})_{t \in [T]}$ as follows:

$$v_t^{(\gamma)} = (1 - \gamma)X_t \theta_t^\star + \gamma \sin^4((2X_t \theta_t^\star)^{-1}) + \varepsilon_t .$$

In Figure 3, we make $\gamma$ vary from 0 to 1. At each value, we plot the average cumulated cost gap between `DF-OGD` and `PF-OGD`, averaged over 50 runs with horizon $T = 1000$. This experiment provides empirical support to the previous conjecture, namely, DFL becomes a competitive option when the prediction model is ill-specified.

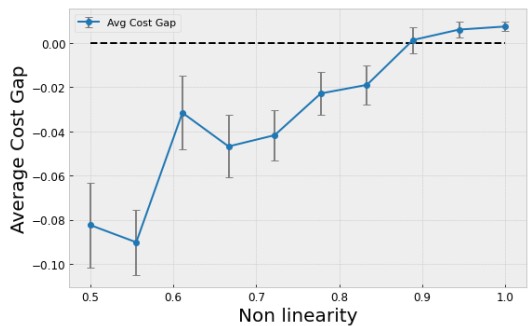

Figure 3: Average cumulated cost gap PFL - DFL as a function of $\gamma$.

---

**Algorithm 4** Online Smart Predict-then-Optimize Online Gradient Descent (Online `SPO`).

---

1: **Input:** horizon $T > 0$, initialization $\theta_1 \in \Theta$.
2: **for** each $t \in \{1, \ldots, T\}$ **do**
3:     Observe $X_t$, predict $\hat{c}_t = g(\theta_t, X_t)$.
4:     Play $v_t^\star(\theta_t) = \operatorname{argmin}_{v \in \mathcal{V}} \hat{c}_t^\mathsf{T} v$, observe $c_t$.
5:     Compute $v^\star(\hat{c}_t) = \operatorname{argmin}_{v \in \mathcal{V}} \hat{c}_t^\mathsf{T} v$ and $v^\star(c_t) = \operatorname{argmin}_{v \in \mathcal{V}} c_t^\mathsf{T} v$.
6:     Define SPO$^+$ surrogate:

$$\ell^{\mathrm{SPO+}}(\hat{c}_t, c_t) = 2c_t^\mathsf{T} v^\star(\hat{c}_t) - c_t^\mathsf{T} v^\star(c_t) - \hat{c}_t^\mathsf{T} v^\star(\hat{c}_t).$$

7:     Update $\theta_{t+1} = \Pi_\Theta\big(\theta_t - \eta \nabla_\theta \ell^{\mathrm{SPO+}}(\hat{c}_t, c_t)\big)$, where the gradient is computed via chain rule through $\hat{c}_t = g(\theta_t, X_t)$.
8: **end for**

---

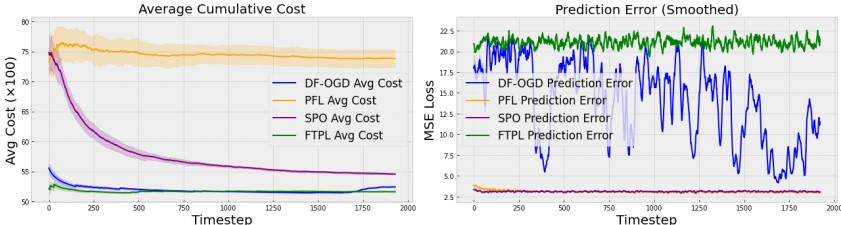

Figure 4: Experiment results with 80 items.

**Higher dimension problem.** Finally, the following figure shows the performance of our algorithms in a higher dimension problem, with $K = 80$ items. We see that our algorithms still significantly beat the baselines, suggesting that they remain effective in moderately high dimension problems.

## F    REGRET BOUND WITH AN APPROXIMATE OPTIMAL SOLUTION.

In this section, we show that when we cannot access $w_t^\star(\theta_t) \in \mathcal{W}$ but only $\underline{w}_t(\theta_t) \in \mathcal{W}$ such that $\langle g(\theta, X_t), \underline{w}_t(\theta_t) - w_t^\star(\theta_t)\rangle \leqslant \kappa$ for $\kappa > 0$, our regret bounds are only shifted by $\kappa$. The following proof is for the general convex polytope case. The simplex case is established in the exact same way.

**Proposition 5.** *Assume the assumptions from Theorem 2 and that at each step, $w_t^\star(\theta_t) \in \mathcal{W}$ is replaced by $\underline{w}_t(\theta_t) \in \mathcal{W}$ which satisfies*

$$\langle g(\theta, X_t), \underline{w}_t(\theta_t) - w_t^\star(\theta_t)\rangle \leqslant \kappa ,$$

*where $\kappa > 0$. Then,*

$$\mathbb{E}\big[\mathfrak{R}_T^d\big] \leqslant \frac{D_\Theta^2 - D_\Theta \mathbb{E}[P_T]}{2\eta_T} + \sum_{t\in[T]} \frac{\eta_t}{2}\left(\frac{GD_\mathcal{Z}\sup_{w\in\mathcal{W}_1}\|b - Aw\|_\infty}{\alpha_t\lambda_{\min}(AA^\intercal)}\right)^2$$
$$+ 2D_\mathcal{Z}\sum_{t\in[T]} \alpha_t[1 + 2n\max_{k\in[K]}\|v_k\|_1 C_0(\ln(1\alpha_t^{-1}) + (1-\beta)^2\ln^2(\alpha_t\ln d))] + \xi + \kappa .$$

The proof Proposition 5 is deferred to Appendix G.6.

## G    PROOFS

### G.1    PROOFS OF PRELIMINARY LEMMAS.

#### G.1.1    SIMPLEX

**Lemma 1.** *Assume H1, $\mathcal{W} = \mathcal{W}_0$ and $\mathcal{R} = \mathcal{R}_0$. Then for any $t \in [T]$, $\tilde{f}_t$ is $K_t$-Lipchitz almost surely, with $K_t = 5D_\mathcal{Z}G(4\alpha_t)^{-1}$.*

*Proof.* Let $(\theta, \theta') \in \Theta^2$. We have:

$$\left|\tilde{f}_t(\theta) - \tilde{f}_t(\theta')\right| = \langle \bar{g}_t(X_t), \tilde{w}_t(\theta) - \tilde{w}_t(\theta')\rangle$$
$$\leqslant \|\bar{g}_t(X_t)\|\|\tilde{w}_t(\theta) - \tilde{w}_t(\theta')\|$$
$$= \|\bar{g}_t(X_t)\| \int_{t=0}^1 \|\nabla\tilde{w}_t(\theta + t(\theta' - \theta))(\theta - \theta')\|\,\mathrm{d}t$$
$$\leqslant D_\mathcal{Z} \int_{t=0}^1 \|\nabla\tilde{w}_t(\theta + t(\theta' - \theta))\|_{\mathrm{op}}\|\theta - \theta'\|\,\mathrm{d}t \quad \text{(by H1-(ii))} \qquad (15)$$

Now, denoting $\zeta = \theta + t(\theta' - \theta)$, we have:

$$
\begin{aligned}
\|\nabla \tilde{w}_t(\zeta)\|_{\mathrm{op}} &= \alpha_t^{-1}\|(\mathrm{diag}[\tilde{w}_t(\zeta)] - \tilde{w}_t(\zeta)\tilde{w}_t(\zeta)^{\mathsf{T}})\nabla_\theta g(\zeta, X_t)\|_{\mathrm{op}} \\
&\leqslant \alpha_t^{-1}\|\mathrm{diag}[\tilde{w}_t(\zeta)] - \tilde{w}_t(\zeta)\tilde{w}_t(\zeta)^{\mathsf{T}}\|_{\mathrm{op}}\|\nabla_\theta g(\zeta, X_t)\|_{\mathrm{op}} \\
&\leqslant G\alpha_t^{-1}\|\mathrm{diag}[\tilde{w}_t(\zeta)] - \tilde{w}_t(\zeta)\tilde{w}_t(\zeta)^{\mathsf{T}}\|_{\mathrm{op}} \quad \text{(by } \mathbf{H}1\text{-(iii))}
\end{aligned} \tag{16}
$$

Since $\mathrm{diag}[\tilde{w}_t(\zeta)] - \tilde{w}_t(\zeta)\tilde{w}_t(\zeta)^{\mathsf{T}}$ is symmetric, its operator norm equals its largest eigenvalue, denoted $\lambda_{\max}$. It then follows that:

$$
\begin{aligned}
\|\mathrm{diag}[\tilde{w}_t(\zeta)] - \tilde{w}_t(\zeta)\tilde{w}_t(\zeta)^{\mathsf{T}}\|_{\mathrm{op}} &= \lambda_{\max}(\mathrm{diag}[\tilde{w}_t(\zeta)] - \tilde{w}_t(\zeta)\tilde{w}_t(\zeta)^{\mathsf{T}}) \\
&\leqslant \max_{i \in [d]} \sum_{j \in [d]} \left|\mathrm{diag}[\tilde{w}_t(\zeta)]_{i,j} - \tilde{w}_t(\zeta)\tilde{w}_t(\zeta)^{\mathsf{T}}_{i,j}\right| \\
&\leqslant \max_{i \in [d]}\{\tilde{w}_t(\zeta)_{i,i}(1 - \tilde{w}_t(\zeta)_{i,i}) + \tilde{w}_t(\theta)_{i,i} \sum_{j \neq i} \tilde{w}_t(\zeta)_{i,j}\} \\
&\leqslant \frac{1}{4} + 1 = \frac{5}{4} \ ,
\end{aligned}
$$

where the two last inequalities hold because $\tilde{w}_t \in [0,1]^d$. Therefore, we obtain from (16) that $\|\nabla \tilde{w}_t(\zeta)\|_{\mathrm{op}} \leqslant 5G(4\alpha_t)^{-1}$, and plugging this in (15) yields:

$$
\left|\tilde{f}_t(\theta) - \tilde{f}_t(\theta')\right| \leqslant \frac{5GD_{\mathcal{Z}}}{4\alpha_t} \int_0^1 \|\theta - \theta'\| \, \mathrm{d}t = \frac{5GD_{\mathcal{Z}}}{4\alpha_t}\|\theta - \theta'\| \ .
$$

$\square$

**Lemma 2.** *Assume* **H**2, $\mathcal{W} = \mathcal{W}_0$ *and* $\mathcal{R} = \mathcal{R}_0$. *Then for any* $\theta \in \Theta$ *and* $t \in [T]$, *for any* $\theta \in \theta$

$$
\mathbb{E}[\|\tilde{w}_t(\theta) - w_t^\star(\theta)\|_1 \mid \mathcal{H}_{t-1}] \leqslant \alpha_t + 2(\alpha_t \ln(d))^\beta C_0 \min\left\{\ln\left(\frac{2}{\alpha_t}\right), (1-\beta)^{-1}\right\} .
$$

*Proof.* Let $\theta \in \Theta$ and $t \in [T]$. Since $\mathcal{W} = \mathcal{W}_0$, denoting for any $j \in [d]$ $e_j$ the $j$-th element of the canonical basis, we have $u_j(\theta, X_t) = \langle g(\theta, X_t), e_j \rangle$. We recall that we denote $u_{I_t(\theta)}(\theta, X_t) = \min_{j \in [d]} u_j(\theta, X_t)$. We first prove the following lemma.

**Lemma 3.** *Assume that there exists* $\varepsilon > 0$ *such that* $u_{I_t(\theta)}(\theta, X_t) + \varepsilon \leqslant u_j(\theta, X_t)$ *for any* $j \in \{1, \ldots, K\} \setminus \{I_t(\theta)\}$. *Then,*

$$
\|w_t^\star(\theta) - \tilde{w}_t(\theta)\|_1 \leqslant \frac{2\alpha_t \ln(d)}{\varepsilon} \ .
$$

*Proof.* In this proof, $I_t(\theta)$ from **H**2 is denoted $I_t$ to lighten notation. We have by assumption:

$$
\begin{aligned}
\langle g(\theta, X_t), \tilde{w}_t(\theta) \rangle &= \left\langle g(\theta, X_t), \sum_{j=1}^d \tilde{w}_{t,j}(\theta) \, e_j \right\rangle \\
&= u_{I_t}(\theta, X_t)\tilde{w}_{t,I_t}(\theta) + \sum_{j \neq I_t} u_j(\theta, X_t)\tilde{w}_{t,j}(\theta) \\
&\geqslant u_{I_t}(\theta, X_t)\tilde{w}_{t,I_t}(\theta) + (u_{I_t}(\theta, X_t) + \varepsilon) \sum_{j \neq I_t} \tilde{w}_{t,j}(\theta) \\
&= u_{I_t}(\theta, X_t)\tilde{w}_{t,I_t}(\theta) + (u_{I_t}(\theta, X_t) + \varepsilon)(1 - \tilde{w}_{t,I_t}(\theta)) \ .
\end{aligned} \tag{17}
$$

The last equality holds because $\|\tilde{w}_t(\theta)\|_1 = 1$. On the other hand, using the facts that $\mathcal{R}_0 \geqslant 0$, $\tilde{w}_t(\theta) = \mathrm{argmin}\langle g(\theta, X_t), w \rangle + \alpha_t \mathcal{R}_0(w)$ and $u_{I_t}(\theta, X_t) = \langle g(\theta, X_t), w_t^\star(\theta) \rangle$ yields:

$$
\begin{aligned}
\langle g(\theta, X_t), \tilde{w}_t(\theta) \rangle &\leqslant \langle g(\theta, X_t), \tilde{w}_t(\theta) \rangle + \alpha_t \mathcal{R}_0(\tilde{w}_t(\theta)) \\
&\leqslant \langle g(\theta, X_t), w_t^\star(\theta) \rangle + \alpha_t \mathcal{R}_0(w_t^\star(\theta)) \\
&= u_{I_t}(\theta, X_t) + \alpha_t \mathcal{R}_0(w_t^\star(\theta)) \\
&\leqslant u_{I_t}(\theta, X_t) + \alpha_t \ln(d).
\end{aligned} \tag{18}
$$

The last line holds because $\mathcal{R}_0(w) \leqslant \ln(d)$ for any $w \in \mathcal{W}_0$. Combining (17) and (18) gives:

$$u_{I_t}(\theta, X_t)\tilde{w}_{t,I_t}(\theta) + (u_{I_t} + \varepsilon)(1 - \tilde{w}_{t,I_t}(\theta)) \leqslant u_{I_t}(\theta, X_t) + \alpha_t \ln(d).$$

Re-organizing the terms then yields $\tilde{w}_{t,I_t}(\theta) \geqslant 1 - \frac{\alpha_t \ln(d)}{\varepsilon}$ and again, as $\|\tilde{w}_t(\theta)\|_1 = 1$, we have: $\sum_{j \neq I_t} \tilde{w}_{t,j}(\theta) \leqslant \frac{\alpha_t \ln(d)}{\varepsilon}$.

Combining those inequalities yields:

$$\|w_t^\star(\theta) - \tilde{w}_t(\theta)\|_1 = \|e_{I_t} - \tilde{w}_t(\theta)\|_1 = 1 - \tilde{w}_{t,I_t}(\theta) + \sum_{j \neq I_t} \tilde{w}_{t,j}(\theta) \leqslant \frac{2\alpha_t \ln(d)}{\varepsilon}.$$

$\square$

To conclude the proof, we first write the expected distance as follows:

$$\mathbb{E}[\|w_t^\star(\theta) - \tilde{w}_t(\theta)\|_1 \mid \mathcal{H}_{t-1}] = \int_0^{+\infty} \mathbb{P}(\|w_t^\star(\theta) - \tilde{w}_t(\theta)\|_1 > y \mid \mathcal{H}_{t-1}) \, \mathrm{d}y,$$

Since both $w_t^\star(\theta)$ and $\tilde{w}_t(\theta)$ belong to the simplex, we can restrict the integral to:

$$= \int_0^2 \mathbb{P}(\|w_t^\star(\theta) - \tilde{w}_t(\theta)\|_1 > y \mid \mathcal{H}_{t-1}) \, \mathrm{d}y \tag{19}$$

$$= \int_0^{\alpha_t} \mathbb{P}(\|w_t^\star(\theta) - \tilde{w}_t(\theta)\|_1 > y \mid \mathcal{H}_{t-1}) \, \mathrm{d}y \tag{20}$$

$$+ \int_{\alpha_t}^2 \mathbb{P}(\|w_t^\star(\theta) - \tilde{w}_t(\theta)\|_1 > y \mid \mathcal{H}_{t-1}) \, \mathrm{d}y$$

$$\leqslant \alpha_t + \int_{\alpha_t}^2 \mathbb{P}(\|w_t^\star(\theta) - \tilde{w}_t(\theta)\|_1 > y \mid \mathcal{H}_{t-1}) \, \mathrm{d}y \tag{21}$$

We now apply the change of variable $y = 2\alpha_t \ln(d)\varepsilon^{-1}$ to obtain:

$$\mathbb{E}[\|w_t^\star(\theta) - \tilde{w}_t(\theta)\|_1 \mid \mathcal{H}_{t-1}]$$
$$= \alpha_t + 2\alpha_t \ln(d) \int_{(\alpha_t \ln(d))}^{(2\ln(d))} \mathbb{P}\left(\|w_t^\star(\theta) - \tilde{w}_t(\theta)\|_1 > 2\frac{\alpha_t \ln(d)}{\varepsilon} \mid \mathcal{H}_{t-1}\right) \frac{\mathrm{d}\varepsilon}{\varepsilon^2}. \tag{22}$$

Moreover, for any $\varepsilon > 0$ we have by **H2** and Lemma 3 that:

$$1 - C_0\varepsilon^\beta \leqslant \mathbb{P}\left(\inf_{j \neq I_t}\{u_j(\theta, X_t) - u_{I_t}(\theta, X_t)\} \geqslant \varepsilon \mid \mathcal{H}_{t-1}\right)$$
$$\leqslant \mathbb{P}\left(\|w_t^\star(\theta) - \tilde{w}_t(\theta)\|_1 \leqslant 2\alpha_t \ln(d)\varepsilon^{-1} \mid \mathcal{H}_{t-1}\right),$$

so it follows that:

$$\mathbb{P}_t\left(\|w_t^\star(\theta) - \tilde{w}_t(\theta)\|_1 > 2\frac{\alpha_t \ln(d)}{\varepsilon} \mid \mathcal{H}_{t-1}\right) = 1 - \mathbb{P}_t\left(\|w_t^\star(\theta) - \tilde{w}_t(\theta)\|_1 \leqslant 2\frac{\alpha_t \ln(d)}{\varepsilon} \mid \mathcal{H}_{t-1}\right)$$
$$\leqslant C_0\varepsilon^\beta, \tag{23}$$

Hence, plugging (23) in (22) gives:

$$\mathbb{E}[\|w_t^\star(\theta) - \tilde{w}_t(\theta)\|_1 \mid \mathcal{H}_{t-1}] \leqslant \alpha_t + 2\alpha_t \ln(d)C_0 \int_{\alpha_t \ln(d)}^{2\ln(d)} \varepsilon^{\beta-2} \, \mathrm{d}\varepsilon.$$

We now proceed to bound $\varphi(\beta) = \int_{\alpha_t \ln(d)}^{2\ln(d)} \varepsilon^{\beta-2} \, \mathrm{d}\varepsilon$ for $\beta \in (0, 1)$. With $a = \alpha_t \ln(d)$ and $b = 2\ln(d)$, we have

$$\varphi(\beta) = (1-\beta)^{-1}(a^{\beta-1} - b^{\beta-1}) = (1-\beta)^{-1}\left[e^{-(1-\beta)\ln(a)} - e^{-(1-\beta)\ln(b)}\right].$$

Factoring out $e^{-(1-\beta)\ln(a)} = a^{\beta-1}$ yields

$$\varphi(\beta) = a^{\beta-1}(1-\beta)^{-1}\left[1 - e^{-(1-\beta)\ln(b/a)}\right] .$$

Since for any $x \in \mathbb{R}$ one has $1 - e^{-x} \leqslant \min\{x, 1\}$, we obtain

$$\varphi(\beta) \leqslant a^{\beta-1}(1-\beta)^{-1}\min\Big\{(1-\beta)\ln(b/a), 1\Big\} = a^{\beta-1}\min\Big\{\ln(b/a), (1-\beta)^{-1}\Big\} .$$

Substituting the values of $a$ and $b$ finally gives

$$\varphi(\beta) \leqslant (\alpha_t \ln d)^{\beta-1}\min\Big\{\ln\Big(\frac{2}{\alpha_t}\Big), (1-\beta)^{-1}\Big\} ,$$

which concludes the proof.

$\square$

### G.1.2 GENERAL POLYTOPE

In this section, we denote by $\mathcal{W}_1$ the general polytope described in **??** and $\mathcal{R}_1$ the log-barrier regularization described in Equation (7).

**Lemma 4.** *Assume **H1**, $\mathcal{W} = \mathcal{W}_1$ and $\mathcal{R} = \mathcal{R}_1$. Then for any $t \in [T]$, $\tilde{f}_t$ is differentiable and for any $\theta \in \Theta$, $\nabla \tilde{f}_t(\theta) = \nabla \tilde{w}_t(\theta) \bar{g}_t(X_t)$ where:*

$$\nabla \tilde{w}_t(\theta) = -\alpha_t^{-1}\left(\sum_{i=1}^n (b_i - A_i^\intercal \tilde{w}_t(\theta))^{-2} A_i A_i^\intercal\right)\nabla_\theta g(\theta, X_t) .$$

*Proof.* Let $\theta \in \mathbb{R}$ and $t \in [T]$. With

$$h_t : (w, \theta) \mapsto \langle g(\theta, X_t), w\rangle + \alpha_t \mathcal{R}_1(w) ,$$

we have $h_t(w) \to +\infty$ as $w \to \text{bdry}(\mathcal{W}_1)$. Since $h_t(\tilde{w}_t(\theta), \theta) = \min_{w \in \mathcal{W}_1} h_t(w, \theta)$, we deduce that $\tilde{w}_t(\theta) \in \text{int}(\mathcal{W}_1)$. It follows from the first order condition and the implicit function theorem (de Oliveira, 2014, Theorem 2) that $\tilde{w}_t : \theta \mapsto \tilde{w}_t(\theta)$ is differentiable, and

$$\nabla \tilde{w}_t(\theta) = -(\nabla_{ww}^2 h_t(\tilde{w}_t(\theta), \theta))^{-1}\nabla_{\theta w}^2 h_t(\tilde{w}_t(\theta), \theta)$$
$$= -\frac{1}{\alpha_t}[\nabla^2 \mathcal{R}_1(\tilde{w}_t(\theta))]^{-1}\nabla_\theta g(\theta, X_t) . \tag{24}$$

Since $\mathcal{R}_1(w) = -\sum_{i=1}^n \ln(b_i - A_i w)$, simple computations give:

$$\nabla^2 \mathcal{R}_1(\tilde{w}_t(\theta)) = \sum_{i=1}^n (b_i - A_i^\intercal w_t(\theta))^{-2} A_i A_i^\intercal , \tag{25}$$

Moreover, since $\text{rank}(AA^\intercal) = d$ by assumption, $\nabla^2 \mathcal{R}_1(\tilde{w}_t(\theta))$ is invertible. Plugging (25) in (24) gives the result. $\square$

**Lemma 5.** *Assume **H1** and $\mathcal{R} = \mathcal{R}_1$. For any $t \in [T]$, $\tilde{f}_t$ is $K_t$-Lipschitz almost-surely, with*

$$K_t = \frac{G D_\mathcal{Z}}{\alpha_t}\frac{\sup_{w \in \mathcal{W}_1}\|b - Aw\|_\infty}{\lambda_{\min}(AA^\intercal)} .$$

*Proof.* Let $(\theta, \theta') \in \Theta^2$. We proved in (15) that:

$$\left|\tilde{f}_t(\theta) - \tilde{f}_t(\theta')\right| \leqslant D_\mathcal{Z}\int_{t=0}^1 \|\nabla \tilde{w}_t(\theta + t(\theta' - \theta))\|_{\text{op}}\|\theta - \theta'\|\, \mathrm{d}t .$$

Denoting $\zeta = \theta + t(\theta' - \theta)$, by (24), since $\|\cdot\|_{\text{op}}$ is sub-multiplicative:

$$\|\nabla \tilde{w}_t(\zeta)\|_{\text{op}} = -\alpha_t^{-1}\left\|[\nabla^2 \mathcal{R}_1(\tilde{w}_t(\zeta)]^{-1}\nabla_\theta g(\theta, X_t)\right\|_{\text{op}}$$
$$\leqslant -\alpha_t^{-1}\left\|[\nabla^2 \mathcal{R}_1(\tilde{w}_t(\zeta)]^{-1}\right\|_{\text{op}} G \quad (\text{by } \mathbf{H}1\text{-(iii)}) .$$

Moreover, we have:

$$\left\|[\nabla^2\mathcal{R}_1(\zeta)]^{-1}\right\|_{\mathrm{op}} = \left\|\left(\sum_{i=1}^n (b_i - A_i^\intercal\tilde{w}_t(\zeta))^{-2}A_iA_i^\intercal\right)^{-1}\right\|_{\mathrm{op}}$$

$$\leqslant \sup_{w\in\mathcal{W}_1}\|b - Aw\|_\infty^2\left\|(AA^\intercal)^{-1}\right\|_{\mathrm{op}} \, ,$$

Since $AA^\intercal$ is symmetric, $\left\|(AA^\intercal)^{-1}\right\|_{\mathrm{op}} = \lambda_{\min}(AA^\intercal)^{-1}$, where $\lambda_{\min}(M)$ is the lowest eigenvalue of $M$. We therefore obtain by (15) and the previous inequalities that almost-surely:

$$\left|\tilde{f}_t(\theta) - \tilde{f}_t(\theta')\right| \leqslant \frac{GD_{\mathcal{Z}}}{\alpha_t}\frac{\sup_{w\in\mathcal{W}_1}\|b - Aw\|_\infty^2}{\lambda_{\min}(AA^\intercal)}\int_{t=0}^1\|\theta - \theta'\|\,\mathrm{d}t$$

$$= \frac{GD_{\mathcal{Z}}}{\alpha_t}\frac{\sup_{w\in\mathcal{W}_1}\|b - Aw\|_\infty^2}{\lambda_{\min}(AA^\intercal)}\|\theta - \theta'\| \, .$$

$\square$

**Lemma 6.** *Assume H2, $\mathcal{W} = \mathcal{W}_1$ and $\mathcal{R} = \mathcal{R}_1$. Then for any $t \in [T]$ and $\theta \in \Theta$,*

$$\mathbb{E}[\|\tilde{w}_t(\theta) - w_t^\star(\theta)\|_1 \mid \mathcal{H}_{t-1}] \leqslant \alpha_t + 2n\max_{k\in[K]}\|v_k\|_1\alpha_t(\alpha_t\ln d)^{\beta-1}\min\left\{\ln\left(\frac{2}{\alpha_t}\right), (1-\beta)^{-1}\right\}$$

*Proof.* Let $\theta \in \Theta$ and $t \in [T]$. Since $\mathcal{W}_1 = \mathrm{Conv}(v_1, \ldots, v_K)$, there exists $(\lambda_{t,I_t}(\theta), \ldots, \lambda_{t,I_t}(\theta)) \in [0,1]^K$ such that

$$\tilde{w}_t(\theta) = \sum_{i=1}^K \lambda_{t,i}(\theta)v_i \quad \text{and} \quad \sum_{i=1}^K \lambda_{t,i}(\theta) = 1 \, .$$

We recall that $u_j(\theta, X_t) = \langle g(\theta, X_t), v_j\rangle$ denotes the value of the objective function on the vertex $v_j \in \mathcal{W}$, and $u_{I_t}(\theta, X_t) = \min_{j\in[K]}u_j(\theta, X_t)$ so $v_{I_t} = w_t^\star(\theta)$. We start by proving the following lemma:

**Lemma 7.** *If there exists $\varepsilon > 0$ such that $u_{I_t} + \varepsilon \leqslant u_j$ for any $j \in \{1, \ldots, K\} \setminus \{I_t\}$, then,*

$$\|w_t^\star(\theta) - \tilde{w}_t(\theta)\|_1 \leqslant \frac{2n\alpha_t}{\varepsilon}\max_{k\in[K]}\|v_k\|_1 \, .$$

*Proof.* On the one hand, $\tilde{w}_t(\theta)$ is by definition the solution to the problem:

$$\tilde{w}_t(\theta) = \underset{w\in\mathcal{W}}{\mathrm{argmin}}\langle g(\theta, X_t), w\rangle - \alpha_t\sum_{i=1}^d\ln(b_i - A_iw) \, , \tag{26}$$

which is uniquely determined by strong convexity of the objective. We know by (Boyd, 2004, page 566) that:

$$\langle g(\theta, X_t), \tilde{w}_t(\theta)\rangle - \inf_{w\in\mathcal{W}_1}\langle g(\theta, X_t), w\rangle \leqslant n\alpha_t \, ,$$

that is

$$\langle g(\theta, X_t), \tilde{w}_t(\theta)\rangle \leqslant u_{I_t} + n\alpha_t \, . \tag{27}$$

On the other hand, by assumption we know that for any $j \neq I_t$, $u_j \geqslant u_{I_t} + \varepsilon$ so:

$$\langle g(\theta, X_t), \tilde{w}_t(\theta)\rangle = \left\langle g(\theta, X_t), \sum_{j=1}^K \lambda_{t,j}(\theta)v_j\right\rangle \tag{28}$$

$$= \lambda_{t,I_t}(\theta)u_{I_t} + \sum_{j\neq I_t}\lambda_{t,j}(\theta)u_j$$

$$\geqslant \lambda_{t,I_t}(\theta)u_{I_t} + \sum_{j\neq I_t}\lambda_{t,j}(\theta)(u_{I_t} + \varepsilon)$$

$$= \lambda_{t,I_t}(\theta)u_{I_t} + (u_{I_t} + \varepsilon)(1 - \lambda_{t,I_t}(\theta)) \, . \tag{29}$$

Combining (27) and (29) yields:

$$\lambda_{t,I_t}(\theta) \geqslant 1 - \frac{n\alpha_t}{\varepsilon} \ , \tag{30}$$

and it follows from (30) that:

$$
\begin{aligned}
\|w_t^\star(\theta) - \tilde{w}_t(\theta)\|_1 = \|v_{I_t} - \tilde{w}_t(\theta)\|_1 &= \left\| \sum_{j=1}^{K} (\mathbb{1}_{\{j=I_t\}} - \lambda_{t,j}(\theta)) v_j \right\|_1 \\
&\leqslant \left[ (1 - \lambda_{t,I_t}(\theta)) + \sum_{j \neq I_t} \lambda_{t,j}(\theta) \right] \max_{k \in [K]} \|v_k\|_1 \\
&\leqslant \frac{2n\alpha_t}{\varepsilon} \max_{k \in [K]} \|v_k\|_1 \ .
\end{aligned}
$$

□

The result follows from combining **H**2 and Lemma 7 according to the same lines of computation as in the proof of Lemma 2. □

### G.1.3 OTHER LEMMAS

**Lemma 8.** *Assume that for any $t \in [T]$, $\tilde{f}_t$ is $K_t-$Lipschitz almost-surely and that the sequence of steps $(\eta_t)_{t \geqslant 1}$ is non-increasing. Denote $\tilde{\mathfrak{R}}_T^d = \sum_{t \in [T]} \tilde{f}_t(\theta_t) - \tilde{f}_t(\vartheta_t)$. Then it holds almost surely that:*

$$\mathbb{E}\left[\tilde{\mathfrak{R}}^d{}_T\right] \leqslant \frac{1}{2\eta_T}\left( D_\Theta^2 + 2D_\Theta \sum_{t=1}^{T} \|\vartheta_{t+1} - \vartheta_t\| \right) + \sum_{t=1}^{T} \frac{\eta_t}{2} K_t^2 \ ,$$

*where the expectation is taken over the sequence $(u_1, \ldots, u_T)$ in Algorithm 2.*

*Proof.* In what follows, we write $\mathbb{E}_{u_t}$ the expectation under the distribution $\mathrm{Unif}([0,1])$ of $u_t$, and $\mathbb{E}_{u_{1:T}}$ the expectation under the joint distribution $\mathrm{Unif}([0,1])^{\otimes T}$ of $(u_1, \ldots, u_T)$. For any $t \in [T]$, we have:

$$
\begin{aligned}
\sum_{t=1}^{T} \tilde{f}_t(\theta_t) - \tilde{f}_t(\vartheta_t) &= \sum_{t=1}^{T} \int_{u=0}^{1} \left\langle \nabla \tilde{f}_t(\vartheta_t + u(\theta_t - \vartheta_t)), \theta_t - \vartheta_t \right\rangle du \\
&= \sum_{t=1}^{T} \left\langle \mathbb{E}_{u_t}\left[ \nabla \tilde{f}_t(\vartheta_t + u_t(\theta_t - \vartheta_t)) \right], \theta_t - \vartheta_t \right\rangle \\
&= \sum_{t=1}^{T} \left\langle \mathbb{E}_{u_t}\left[ \tilde{\nabla}_t(u_t) \right], \theta_t - \vartheta_t \right\rangle \ , \tag{31}
\end{aligned}
$$

where $\tilde{\nabla}(u_t) = \nabla \tilde{f}_t(\vartheta_t + u_t(\theta_t - \vartheta_t))$ as in Algorithm 2. To control (31) we note that by definition of $\theta_{t+1}$ in Algorithm 2,

$$
\begin{aligned}
\mathbb{E}_{u_t}\left[ \|\theta_{t+1} - \vartheta_t\|^2 \right] = \mathbb{E}_{u_t}\left[ \left\| \Pi_\Theta[\theta_t - \eta_t \tilde{\nabla}_t(u_t)] - \vartheta_t \right\|^2 \right] \\
\leqslant \mathbb{E}_{u_t}\left[ \left\| \theta_t - \vartheta_t - \eta_t \tilde{\nabla}_t(u_t) \right\|^2 \right] \\
\leqslant \|\theta_t - \vartheta_t\|^2 - 2\eta_t \left\langle \mathbb{E}_{u_t}\left[ \tilde{\nabla}_t(u_t) \right], \theta_t - \vartheta_t \right\rangle + \eta_t^2 K_t^2 \tag{32}
\end{aligned}
$$

where we have used that $\left\| \tilde{\nabla}_t(u_t) \right\|^2 \leqslant K_t^2$ because $\tilde{f}_t$ is $K_t-$Lipschitz. Re-arranging (32) and taking the expectation over $(u_1, \ldots, u_T)$ yields:

$$\mathbb{E}_{u_{1:T}}\left[ \left\langle \tilde{\nabla}_t(u_t), \theta_t - \vartheta_t \right\rangle \right] \leqslant \mathbb{E}_{u_{1:T}}\left[ \frac{1}{2\eta_t}\left( \|\theta_t - \vartheta_t\|^2 - \|\theta_{t+1} - \vartheta_t\|^2 \right) + \frac{\eta_t K_t^2}{2} \right] \ , \tag{33}$$

Now notice that, for any $t \in \{1, \ldots, T-1\}$,

$$
\begin{aligned}
\|\theta_{t+1} - \vartheta_{t+1}\|^2 &= \|\theta_{t+1} - \vartheta_t\|^2 + \|\theta_{t+1} - \vartheta_{t+1}\|^2 - \|\theta_{t+1} - \vartheta_t\|^2 \\
&= \|\theta_{t+1} - \vartheta_t\|^2 \\
&\quad + (\|\theta_{t+1} - \vartheta_{t+1}\| + \|\theta_{t+1} - \vartheta_t\|)(\|\theta_{t+1} - \vartheta_{t+1}\| - \|\theta_{t+1} - \vartheta_t\|) \\
&\leqslant \|\theta_{t+1} - \vartheta_t\|^2 + 2 D_\Theta \|\vartheta_{t+1} - \vartheta_t\|,
\end{aligned}
\tag{34}
$$

where we have used **H**1-(i) and the reversed triangular inequality in the last line. It follows that:

$$
-\|\theta_{t+1} - \vartheta_t\|^2 \leqslant -\|\theta_{t+1} - \vartheta_{t+1}\|^2 + 2 D_\Theta \|\vartheta_{t+1} - \vartheta_t\|
\tag{35}
$$

Plugging (35) into (33) then gives:

$$
\begin{aligned}
\mathbb{E}_{u_{1:T}}\left[\left\langle \tilde{\nabla}_t(u_t), \theta_t - \vartheta_t \right\rangle\right] &\leqslant \mathbb{E}_{u_{1:T}}\left[\frac{1}{2\eta_t}\left(\|\theta_t - \vartheta_t\|^2 - \|\theta_{t+1} - \vartheta_{t+1}\|^2\right)\right. \\
&\quad \left. + \frac{1}{\eta_t} D_\theta \|\vartheta_{t+1} - \vartheta_t\| + \frac{\eta_t K_t^2}{2}\right].
\end{aligned}
$$

Thus, summing over $T$ and plugging the resulting sum in (31) gives:

$$
\begin{aligned}
\mathbb{E}_{u_{1:T}}\left[\sum_{t=1}^T \tilde{f}_t(\theta_t) - \tilde{f}_t(\vartheta_t)\right] &\leqslant \mathbb{E}_{u_{1:T}}\left[\sum_{t=1}^T \frac{\|\theta_t - \vartheta_t\|^2}{2}\left(\frac{1}{\eta_t} - \frac{1}{\eta_{t-1}}\right)\right. \\
&\quad \left. + \sum_{t=1}^T \frac{1}{\eta_t} D_\theta \|\vartheta_{t+1} - \vartheta_t\| + \sum_{t=1}^T \frac{\eta_t K_t^2}{2}\right]. \\
&\leqslant \frac{D_\Theta^2}{2}\sum_{t=1}^T\left(\frac{1}{\eta_t} - \frac{1}{\eta_{t-1}}\right) + \frac{D_\theta}{\eta_T}\sum_{t=1}^T \|\vartheta_{t+1} - \vartheta_t\| + \sum_{t=1}^T \frac{\eta_t K_t^2}{2}.
\end{aligned}
$$

where we used that for all $t$, $\|\theta_t - \vartheta_t\| \leqslant D_\Theta$ by **H**1-(i) and that for all $t \in [T]$, $\eta_{t-1} \geqslant \eta_t \geqslant \eta_T$. Then, we have by telescoping the first sum:

$$
\mathbb{E}_{u_{1:T}}\left[\sum_{t=1}^T \tilde{f}_t(\theta_t) - \tilde{f}_t(\vartheta_t)\right] \leqslant \frac{D_\Theta^2 + 2 D_\Theta}{2\eta_T}\sum_{t=1}^T \|\vartheta_{t+1} - \vartheta_t\| + \sum_{t=1}^T \frac{\eta_t K_t^2}{2}.
$$

$\square$

## G.2 Proof of Theorem 1.

**Theorem 1.** *Assume **H**1, **H**2 and having access to an $\xi$-approximate oracle $\mathbf{O}_\xi$ adapted to $\{\sum_{i=1}^t \tilde{f}_i - \langle \sigma_t, \cdot \rangle\}_{t \in [T]}$. Let $\{\theta_t\}_{t \in [T]}$ be the output of DF-FTPL (Algorithm 1) instantiated with learning step $\eta > 0$ and regularization coefficients $\alpha_t = \alpha > 0$ for any $t$. Then:*

$$
T^{-1}\mathbb{E}[\mathfrak{R}_T^s] = \tilde{\mathcal{O}}\left(\eta m^2 D \frac{1}{\alpha^2} + \frac{mD}{\eta T} + \xi + \alpha n\right),
$$

*where $\mathbb{E}$ denotes the expectation on both data and the intrinsic randomness of DF-FTPL and $\tilde{\mathcal{O}}$ contains polynomial dependency in $\ln(1/\alpha), \ln(\ln(d))$.*
*Furthermore, taking $\eta \propto m^{1/4} T^{-3/4} n^{-1/2}$ and $\alpha \propto m^{3/4} n^{1/2} T^{-1/4}$ yields:*

$$
T^{-1}\mathbb{E}[\mathfrak{R}_T^s] = \tilde{\mathcal{O}}\left(m^{3/4} n^{3/2} T^{-1/4} + \xi\right).
$$

*Proof.* For the sake of conciseness we use the notation $\mathbb{E}_t$ to denote $\mathbb{E}[\cdot \mid \mathcal{H}_{t-1}]$ In what follows, we consider, for any $\theta$, the intermediary regret: $\mathrm{R}_T^s(\theta) := \sum_{t=1}^T F_t(\theta_t) - F_t(\theta)$, where $\quad F_t : \theta \mapsto$

$\mathbb{E}[f_t(\theta) \mid \mathcal{H}_{t-1}]$. We have, for any $\theta$, the following decomposition of the static regret

$$R_T^s(\theta) = \sum_{t \in [T]} \mathbb{E}_t[\langle \bar{g}_t(X_t), w_t^\star(\theta_t) - \tilde{w}_t(\theta_t) \rangle] \tag{36}$$

$$+ \sum_{t \in [T]} \mathbb{E}_t[\langle \bar{g}_t(X_t), \tilde{w}_t(\theta_t) \rangle] - \sum_{t \in [T]} \mathbb{E}_t[\langle \bar{g}_t(X_t), \tilde{w}_t(\theta) \rangle] \tag{37}$$

$$+ \sum_{t \in [T]} \mathbb{E}_t[\langle \bar{g}_t(X_t), \tilde{w}_t(\theta) \rangle] - \sum_{t \in [T]} \mathbb{E}_t[\langle \bar{g}_t(X_t), w_t^\star(\theta) \rangle] \tag{38}$$

Then, taking the sup over $\Theta$ in Equations (36) and (38), and defining

$$\text{Reg}_T^S := \sum_{t \in [T]} \mathbb{E}_t[\langle \bar{g}_t(X_t), \tilde{w}_t(\theta_t) \rangle] - \sum_{t \in [T]} \mathbb{E}_t[\langle \bar{g}_t(X_t), \tilde{w}_t(\theta) \rangle],$$

in Equation (37) yields for any $\theta$:

$$R_T^s(\theta) \leqslant \text{Reg}_T^S + 2 \sum_{t \in [T]} \sup_{\theta \in \Theta} \mathbb{E}_t[\langle \bar{g}_t(X_t), w_t^\star(\theta) - \tilde{w}_t(\theta) \rangle]$$

Using the fact that $\|\cdot\|_\infty$ is dual to $\|\cdot\|_1$:

$$R_T^s(\theta) \leqslant \text{Reg}_T^S + 2 \sum_{t \in [T]} \sup_{\theta \in \Theta} \mathbb{E}_t[\|\bar{g}_t(X_t)\|_\infty \|w_t^\star(\theta) - \tilde{w}_t(\theta)\|_1]$$

Since for any $t \in [T]$, $\|\bar{g}_t(X_t)\|_\infty \leqslant \|\bar{g}_t(X_t)\|_2 \leqslant D_{\mathcal{Z}}$ by **H**1-(ii),

$$R_T^s(\theta) \leqslant \text{Reg}_T^S + 2D_{\mathcal{Z}} \sum_{t \in [T]} \mathbb{E}_t \sup_{\theta \in \Theta} \|w_t^\star(\theta) - \tilde{w}_t(\theta)\|_1$$

First, notice that, by Lemma 6, and because for any $t$, $\alpha_t = \alpha$ we have for all $t$:

$$\mathbb{E}[\|\tilde{w}_t(\theta) - w_t^\star(\theta)\|_1 \mid \mathcal{H}_{t-1}] \leqslant \alpha + 2n \max_{k \in [K]} \|v_k\|_1 \alpha(\alpha \ln d)^{\beta-1} \min\left\{\ln\left(\frac{2}{\alpha}\right), (1-\beta)^{-1}\right\}. \tag{39}$$

Second, taking the expectation (denoted by $\mathbb{E}$) over $(\sigma_t)_{t \in [T]}$ and $X_1, \cdots X_T$ on both sides gives:

$$\mathbb{E}[R_T^s(\theta)] \leqslant \mathbb{E}[\text{Reg}_T^S] + 2D_{\mathcal{Z}}T\left(\alpha + 2n \max_{k \in [K]} \|v_k\|_1 \alpha(\alpha \ln d)^{\beta-1} \min\left\{\ln\left(\frac{2}{\alpha}\right), (1-\beta)^{-1}\right\}\right). \tag{40}$$

We now bound the first term in the right-hand-side of Equation (40). By the definition of conditional expectation, we have:

$$\mathbb{E}[\text{Reg}_T^S] = \mathbb{E}_{X_1 \cdots X_T} \mathbb{E}_{\sigma_1, \cdots, \sigma_T}\left[\sum_{t \in [T]} \langle \bar{g}_t(X_t), \tilde{w}_t(\theta_t) \rangle - \langle \bar{g}_t(X_t), \tilde{w}_t(\theta) \rangle\right]$$

and then,

$$\mathbb{E}[\text{Reg}_T^S] \leqslant \mathbb{E}_{X_1 \cdots X_T} \mathbb{E}_{\sigma_1, \cdots, \sigma_T}\left[\sum_{t \in [T]} \tilde{f}_t(\theta_t) - \inf_{\theta \in \Theta} \sum_{t \in [T]} \tilde{f}_t(\theta)\right].$$

One recognises (up to a factor $T$) the left-hand side of Proposition 4 on the loss sequence $(\tilde{f}_t)_{t \in [T]}$. Furthermore, we can use this proposition as, given our choice of $\mathcal{R}$, for any $t \in [T]$, $\tilde{f}_t$ is $L$−Lipschitz with $L = 5D_{\mathcal{Z}}G(4\alpha)^{-1}$ almost surely by Lemma 1 (with $\chi = 0$). We then have:

$$\mathbb{E}_{\sigma_1, \cdots, \sigma_T}\left[\frac{1}{T} \sum_{t \in [T]} \tilde{f}_t(\theta_t) - \inf_{\theta \in \Theta} \frac{1}{T} \sum_{t \in [T]} \tilde{f}_t(\theta)\right] \leqslant \mathcal{O}\left(\eta m^2 D \frac{1}{\alpha^2} + \frac{mD}{\eta T} + \xi\right). \tag{41}$$

Dividing Equation (40) by $T$, and plugging Equation (41) gives: for all $\theta \in \Theta$:

$$
\begin{aligned}
T^{-1}\mathbb{E}[\mathrm{R}_T^s(\theta)] &\leqslant \mathcal{O}\left(\eta m^2 D \frac{1}{\alpha^2} + \frac{mD}{\eta T} + \xi\right) \\
&\quad + 2D_{\mathcal{Z}}T\left(\alpha + 2n \max_{k\in[K]}\|v_k\|_1 \alpha(\alpha \ln d)^{\beta-1} \min\left\{\ln\left(\frac{2}{\alpha}\right), (1-\beta)^{-1}\right\}\right) \\
&= \tilde{\mathcal{O}}\left(\eta m^2 D \frac{1}{\alpha^2} + \frac{mD}{\eta T} + \xi + \alpha n\right).
\end{aligned}
$$

Finally remark that, by the definition of the conditional expectation (thus of $F_t$), and because $\theta_t$ is $\mathcal{F}_{t-1}$-measurable, we have for any $\theta$:

$$
\begin{aligned}
\mathbb{E}[\mathrm{R}_T^s(\theta)] &= \mathbb{E}\left[\sum_{t=1}^T f_t(\theta_t) - f_t(\theta)\right] \\
&= \mathbb{E}\left[\sum_{t=1}^T f_t(\theta_t)\right] - \mathbb{E}[f_t(\theta)]
\end{aligned}
$$

Then taking the infimum over $\theta$ yields:

$$
T^{-1}\mathbb{E}[\mathfrak{R}_T^s] \leqslant \tilde{\mathcal{O}}\left(\eta m^2 D \frac{1}{\alpha^2} + \frac{mD}{\eta T} + \xi + \alpha n\right).
$$

This concludes the proof. The second equation consists in simply plugging the proposed value of $\eta, \alpha$ in this bound. $\qquad\square$

### G.3 Proof of Theorem 2

**Theorem 2.** *Assume H1, H2, access to a $\xi$-approximate oracle adapted to $\{\tilde{f}_t\}_{t\in[T]}$. Let $\{\theta_t\}_{t\in[T]}$ be the output of* DF-OGD *(Algorithm 2) instantiated with the non-increasing sequence $(\eta_t)_{t\in[T]}$ and regularization coefficients $(\alpha_t)_{t\in[T]}$. Then:*

$$
T^{-1}\mathbb{E}\left[\mathfrak{R}_T^d\right] = \tilde{\mathcal{O}}\left(\mathbb{E}\left[\frac{1+P_T}{T\eta_T} + \frac{1}{T}\sum_{t\in[T]}\frac{\eta_t}{\alpha_t^2} + n\alpha_t\right] + \xi\right).
$$

*where $P_T = \sum_{t=1}^{T-1}\|\vartheta_{t+1} - \vartheta_t\|$, $\mathbb{E}$ denotes the expectation on both data and the intrinsic randomness of* DF-OGD *and $\tilde{\mathcal{O}}$ contains polynomial dependency in $\ln(1/\alpha), \ln(\ln(d))$.*

*Furthermore, assume $(t^{-1}(1 + P_t))_{t\geqslant 1}$ is non-increasing almost surely with $P_t = \sum_{s=1}^{t-1}\|\vartheta_{s+1} - \vartheta_s\|$. Then, using $\alpha_t \propto n^{-1/2}t^{-1/4}(1+P_t)^{1/4}$ and $\eta_t \propto n^{-1/2}t^{-3/4}(1+P_t)^{3/4}$ for any $t \in [T]$ leads to:*

$$
T^{-1}\mathbb{E}\left[\mathfrak{R}_T^d\right] = \tilde{\mathcal{O}}\left(\mathbb{E}\left[\sqrt{n}(1+P_T)^{1/4}T^{-1/4}\right] + \xi\right).
$$

*Proof.* In this proof, for the sake of conciseness, we use the notation $\mathbb{E}_t$ to denote $\mathbb{E}[\cdot \mid \mathcal{H}_{t-1}]$. Observe that the dynamic regret can be decomposed as follows:

$$
\mathfrak{R}_T^d = \sum_{t\in[T]}\mathbb{E}_t[\langle\bar{g}_t(X_t), w_t^\star(\theta_t) - \tilde{w}_t(\theta_t)\rangle] + \sum_{t\in[T]}\mathbb{E}_t[\langle\bar{g}_t(X_t), \tilde{w}_t(\theta_t) - \tilde{w}_t(\vartheta_t)\rangle] \tag{42}
$$

$$
+ \sum_{t\in[T]}\mathbb{E}_t[\langle\bar{g}_t(X_t), \tilde{w}_t(\vartheta_t)\rangle] - \inf_{\theta\in\Theta}\mathbb{E}_t[\langle\bar{g}_t(X_t), \tilde{w}_t(\theta)\rangle] \tag{43}
$$

$$
+ \sum_{t\in[T]}\left[\inf_{\theta\in\Theta}\mathbb{E}_t[\langle\bar{g}_t(X_t), \tilde{w}_t(\theta)\rangle] - \inf_{\theta\in\Theta}\mathbb{E}_t[\langle\bar{g}_t(X_t), w_t^\star(\theta)\rangle]\right] \tag{44}
$$

First, we remark that for any $t$, given $\vartheta_t = \mathbf{O}(\tilde{f}_t)$, we control (43) by Jensen:

$$
\mathbb{E}_t[\langle \bar{g}_t(X_t), \tilde{w}_t(\vartheta_t) \rangle] - \inf_{\theta \in \Theta} \mathbb{E}_t[\langle \bar{g}_t(X_t), \tilde{w}_t(\theta) \rangle] \leqslant \mathbb{E}_t \left[ \tilde{f}_t(\vartheta_t) - \inf_{\theta \in \Theta} \tilde{f}_t(\theta) \right]
$$
$$
\leqslant \xi.
$$

Then, taking the $\sup$ over $\Theta$ for each summand of the first sum of (42) (which is valid as $\theta_t$ is $\mathcal{F}_{t-1}$-measurable), defining by $\mathrm{Reg}_T := \sum_{t \in [T]} \mathbb{E}_t[\langle \bar{g}_t(X_t), \tilde{w}_t(\theta_t) - \tilde{w}_t(\vartheta_t) \rangle]$, and noticing for (44) that $\inf_{\theta \in \Theta} \mathbb{E}_t[f_t(\theta)] - \inf_{\theta \in \Theta} \mathbb{E}_t[\tilde{f}_t(\theta)] \leqslant |\inf_{\theta \in \Theta} \mathbb{E}_t[f_t(\theta)] - \inf_{\theta \in \Theta} \mathbb{E}_t[\tilde{f}_t(\theta)]| \leqslant \sup_{\theta \in \Theta} |\mathbb{E}_t[f_t(\theta) - \tilde{f}_t(\theta)]| \leqslant \sup_{\theta \in \Theta} \mathbb{E}_t[|f_t(\theta) - \tilde{f}_t(\theta)|]$ leads to:

$$
\mathfrak{R}_T^d \leqslant \mathrm{Reg}_T + 2 \sum_{t \in [T]} \sup_{\theta \in \Theta} \mathbb{E}_t[|\langle \bar{g}_t(X_t), w_t^\star(\theta) - \tilde{w}_t(\theta) \rangle|] + \xi T
$$

Using the fact that $\|\cdot\|_\infty$ is dual to $\|\cdot\|_1$:

$$
\mathfrak{R}_T^d \leqslant \mathrm{Reg}_T + 2 \sum_{t \in [T]} \sup_{\theta \in \Theta} \mathbb{E}_t[\|\bar{g}_t(X_t)\|_\infty \|w_t^\star(\theta) - \tilde{w}_t(\theta)\|_1] + \xi T
$$

Since for any $t \in [T]$, $\|\bar{g}_t(X_t)\|_\infty \leqslant \|\bar{g}_t(X_t)\|_2 \leqslant D_{\mathcal{Z}}$ almost surely by **H**1-(ii),

$$
\mathfrak{R}_T^d \leqslant \mathrm{Reg}_T + 2 D_{\mathcal{Z}} \sum_{t \in [T]} \sup_{\theta \in \Theta} \mathbb{E}_t[\|w_t^\star(\theta) - \tilde{w}_t(\theta)\|_1] + \xi T \tag{45}
$$

First, by Lemma 6, we know have for any $t$:

$$
\sup_{\theta \in \Theta} \mathbb{E}_t[\|w_t^\star(\theta) - \tilde{w}_t(\theta)\|_1] \leqslant \alpha_t + 2n \max_{k \in [K]} \|v_k\|_1 \alpha_t (\alpha_t \ln d)^{\beta - 1} \min\left\{ \ln\left(\frac{2}{\alpha_t}\right), (1 - \beta)^{-1} \right\}.
$$

Thus combining the two last equations and taking the expectation (denoted by $\mathbb{E}$) over $(u_t)_{t \in [T]}$ and $X_1, \cdots X_T$ on both sides gives:

$$
\mathbb{E}[\mathfrak{R}_T^d] \leqslant \mathbb{E}[\mathrm{Reg}_T]
$$
$$
+ 2 D_{\mathcal{Z}} \sum_{t \in [T]} \left( \alpha_t + 2n \max_{k \in [K]} \|v_k\|_1 \alpha_t (\alpha_t \ln d)^{\beta - 1} \min\left\{ \ln\left(\frac{2}{\alpha_t}\right), (1 - \beta)^{-1} \right\} \right) + \xi T. \tag{46}
$$

Now, to bound the first term in Equation (46) note that the definition of conditional expectation implies:

$$
\mathbb{E}[\mathrm{Reg}_T] = \mathbb{E}_{X_1 \cdots X_T} \mathbb{E}_{u_1, \cdots, u_T} \left[ \sum_{t \in [T]} \tilde{f}_t(\theta_t) - \tilde{f}_t(\vartheta_t) \right].
$$

One recognizes the definition of $\tilde{\mathfrak{R}}_T^d$ of Lemma 8. Since $\tilde{f}_t$ is $K_t-$Lipschitz with $K_t = \frac{G D_{\mathcal{Z}}}{\alpha_t} \frac{\sup_{w \in \mathcal{W}_1} \|b - Aw\|_\infty}{\lambda_{\min}(AA^\intercal)}$ almost surely for any $t$ by Lemma 5, we deduce from Lemma 8 that:

$$
\mathbb{E}_{u_1, \cdots, u_T} \left[ \sum_{t \in [T]} \tilde{f}_t(\theta_t) - \tilde{f}_t(\vartheta_t) \right] \leqslant \frac{D_\Theta^2 - D_\Theta P_T}{2 \eta_T} + \sum_{t \in [T]} \frac{\eta_t}{2} \left( \frac{G D_{\mathcal{Z}} \sup_{w \in \mathcal{W}_1} \|b - Aw\|_\infty^2}{\alpha_t \lambda_{\min}(AA^\intercal)} \right)^2.
$$
$$
\tag{47}
$$

Plugging Equation (47) into Equation (46) and then dividing by $T > 0$ on both sides gives:

$$T^{-1}\mathbb{E}\big[\mathfrak{R}_T^d\big] \leqslant \frac{D_\Theta^2 + D_\Theta \mathbb{E}[P_T]}{2T\eta_T} + \sum_{t\in[T]} \frac{\eta_t}{2}\left(\frac{GD_\mathcal{Z}\sup_{w\in\mathcal{W}_1}\|b - Aw\|_\infty^2}{\alpha_t \lambda_{\min}(AA^\intercal)}\right)^2$$
$$+ 2D_\mathcal{Z}\sum_{t\in[T]}\left(\alpha_t + 2n\max_{k\in[K]}\|v_k\|_1 \alpha_t(\alpha_t \ln d)^{\beta-1}\min\left\{\ln\left(\frac{2}{\alpha_t}\right), (1-\beta)^{-1}\right\}\right) + \xi\,.$$

$$(48)$$

Rewriting Equation (48) using $\tilde{\mathcal{O}}$ concludes the proof.

Concerning the second bound, We know by Equation (48) that the regret of Algorithm 2 satisfies the following bound:

$$\mathbb{E}\big[\mathfrak{R}_T^d\big] = \tilde{\mathcal{O}}\left(\mathbb{E}\left[\frac{1+P_T}{\eta_T} + \sum_{t\in[T]}\frac{\eta_t}{\alpha_t^2} + n\sum_{t\in[T]}\alpha_t + \xi T\right]\right)\,.$$

With $\alpha_t \propto n^{-1/2}t^{-1/4}(1+P_t)^{1/4}$ and $\eta_t \propto \alpha_t t^{-1/2}(1+P_t)^{1/2} \propto n^{-1/2}t^{-3/4}(1+P_t)^{3/4}$, the sequence $(\eta_t)_{t\in[T]}$ is non-increasing because so is $(t^{-1}(1+P_t))_{t\in[T]}$ by assumption. Moreover, the first term satisfies:

$$\frac{1+P_T}{\eta_T} = \mathcal{O}\left(n^{1/2}\frac{T^{3/4}}{(1+P_T)^{3/4}}(1+P_T)\right) \quad \text{so} \quad \frac{1+P_T}{T\eta_T} = \mathcal{O}\left(n^{1/2}\left(\frac{1+P_T}{T}\right)^{1/4}\right)\,.$$

For the second term,

$$\sum_{t=1}^T \frac{\eta_t}{\alpha_t^2} = \sum_{t=1}^T \frac{\sqrt{1+P_t}}{\sqrt{t}\alpha_t} = \mathcal{O}\left(n^{1/2}\sum_{t=1}^T\left(\frac{1+P_t}{t}\right)^{1/4}\right)\,..$$

Moreover, since $1 + P_t \leqslant 1 + P_T$ for any $t \in [T]$, we have:

$$\sum_{t=1}^T \frac{\eta_t}{\alpha_t^2} = \mathcal{O}\left(n^{1/2}(1+P_T)^{1/4}\sum_{t=1}^T\left(\frac{1}{t}\right)^{1/4}\right) = \mathcal{O}\left(n^{1/2}(1+P_T)^{1/4}T^{3/4}\right)\,,$$

where we used in the last line that $\sum_{t=1}^T t^{-\gamma} = \mathcal{O}(T^{1-\gamma})$ for $\gamma \in (0,1)$. Dividing by $T$ again gives a rate of $\mathcal{O}\left(\left(\frac{1+P_T}{T}\right)^{1/4}\right)$ for the second term. Finally, we obtain by the same reasoning that

$$n\sum_{t\in[T]}\alpha_t \leqslant n^{1/2}(1+P_T)^{1/4}\sum_{t\in[T]}t^{-1/4} = \mathcal{O}\left(n^{1/2}(1+P_T)^{1/4}T^{3/4}\right)\,,$$

and dividing by $T > 0$ yields the desired rate. $\qquad\qquad\square$

### G.4 Proof of Proposition 2

**Proposition 2.** *Assume H1, H2 and having access to an $\xi$-approximate optimization oracle $\mathbf{O}_\xi$ adapted to $\left\{\sum_{i=1}^t \tilde{f}_i - \langle\sigma_t, \cdot\rangle\right\}_{t\in[T]}$. Fix $\mathcal{W} = \mathcal{W}_0$, $\mathcal{R} = \mathcal{R}_0$. Let $\{\theta_t\}_{t\in[T]}$ be the output of* `DF-FTPL` *(Algorithm 1) instantiated with learning step $\eta > 0$ and regularization coefficients $\alpha_t = \alpha > 0$ for any $t$. Then:*

$$T^{-1}\mathbb{E}\big[\mathfrak{R}_T^s\big] \leqslant \tilde{\mathcal{O}}\left(\eta m^2 D\frac{1}{\alpha^2} + \frac{mD}{\eta T} + \xi + \alpha\ln(d)\right)\,,$$

*where $\mathbb{E}$ denotes the expectation on both data and the intrinsic randomness of* `DF-FTPL` *and $\tilde{\mathcal{O}}$ contains polynomial dependency in $\ln(1/\alpha), \ln(\ln(d))$.*

*Proof.* For the sake of conciseness we use the notation $\mathbb{E}_t$ to denote $\mathbb{E}[\cdot \mid \mathcal{H}_{t-1}]$ In what follows, we consider, for any $\theta$, the intermediary regret: $\mathrm{R}_T^s(\theta) := \sum_{t=1}^T F_t(\theta_t) - F_t(\theta)$, where $F_t : \theta \mapsto \mathbb{E}[f_t(\theta) \mid \mathcal{H}_{t-1}]$. We have, for any $\theta$, the following decomposition of the static regret

$$\mathrm{R}_T^s(\theta) = \sum_{t \in [T]} \mathbb{E}_t[\langle \bar{g}_t(X_t), w_t^\star(\theta_t) - \tilde{w}_t(\theta_t)\rangle] \tag{49}$$

$$+ \sum_{t \in [T]} \mathbb{E}_t[\langle \bar{g}_t(X_t), \tilde{w}_t(\theta_t)\rangle] - \sum_{t \in [T]} \mathbb{E}_t[\langle \bar{g}_t(X_t), \tilde{w}_t(\theta)\rangle] \tag{50}$$

$$+ \sum_{t \in [T]} \mathbb{E}_t[\langle \bar{g}_t(X_t), \tilde{w}_t(\theta)\rangle] - \sum_{t \in [T]} \mathbb{E}_t[\langle \bar{g}_t(X_t), w_t^\star(\theta)\rangle] \tag{51}$$

Then, taking absolute values, using the triangular inequality and taking the sup over $\Theta$ in Equations (49) and (51), as well as defining

$$\mathrm{Reg}_T^{\mathrm{S}} := \sum_{t \in [T]} \mathbb{E}_t[\langle \bar{g}_t(X_t), \tilde{w}_t(\theta_t)\rangle] - \sum_{t \in [T]} \mathbb{E}_t[\langle \bar{g}_t(X_t), \tilde{w}_t(\theta)\rangle],$$

in Equation (50) yields for any $\theta$:

$$\mathrm{R}_T^s(\theta) \leqslant \mathrm{Reg}_T^{\mathrm{S}} + 2 \sum_{t \in [T]} \sup_{\theta \in \Theta} \mathbb{E}_t[|\langle \bar{g}_t(X_t), w_t^\star(\theta) - \tilde{w}_t(\theta)\rangle|]$$

Using the fact that $\|\cdot\|_\infty$ is dual to $\|\cdot\|_1$:

$$\mathrm{R}_T^s(\theta) \leqslant \mathrm{Reg}_T^{\mathrm{S}} + 2 \sum_{t \in [T]} \sup_{\theta \in \Theta} \mathbb{E}_t[\|\bar{g}_t(X_t)\|_\infty \|w_t^\star(\theta) - \tilde{w}_t(\theta)\|_1]$$

Since for any $t \in [T]$, $\|\bar{g}_t(X_t)\|_\infty \leqslant \|\bar{g}_t(X_t)\|_2 \leqslant D_{\mathcal{Z}}$ by **H**1-(ii),

$$\mathrm{R}_T^s(\theta) \leqslant \mathrm{Reg}_T^{\mathrm{S}} + 2D_{\mathcal{Z}} \sum_{t \in [T]} \mathbb{E}_t \sup_{\theta \in \Theta} \|w_t^\star(\theta) - \tilde{w}_t(\theta)\|_1$$

First, notice that, by Lemma 2, and because for any $t, \alpha_t = \alpha$ we have for all $t$:

$$\mathbb{E}[\|\tilde{w}_t(\theta) - w_t^\star(\theta)\|_1 \mid \mathcal{H}_{t-1}] \leqslant \alpha + 2(\alpha \ln(d))^\beta C_0 \min\left\{\ln\left(\frac{2}{\alpha}\right), (1-\beta)^{-1}\right\}, . \tag{52}$$

Second, taking the expectation (denoted by $\mathbb{E}$) over $(\sigma_t)_{t \in [T]}$ and $X_1, \cdots X_T$ on both sides gives:

$$\mathbb{E}[\mathrm{R}_T^s(\theta)] \leqslant \mathbb{E}[\mathrm{Reg}_T^{\mathrm{S}}] + 2D_{\mathcal{Z}}T\left(\alpha + 2(\alpha \ln(d))^\beta C_0 \min\left\{\ln\left(\frac{2}{\alpha}\right), (1-\beta)^{-1}\right\}\right). \tag{53}$$

To bound the first term in the right-hand-side of Equation (53). by the definition of conditional expectation, we have:

$$\mathbb{E}[\mathrm{Reg}_T^{\mathrm{S}}] = \mathbb{E}_{X_1 \cdots X_T} \mathbb{E}_{\sigma_1, \cdots, \sigma_T}\left[\sum_{t \in [T]} \langle \bar{g}_t(X_t), \tilde{w}_t(\theta_t)\rangle - \langle \bar{g}_t(X_t), \tilde{w}_t(\theta)\rangle\right]$$

Then,

$$\mathbb{E}[\mathrm{Reg}_T^{\mathrm{S}}] \leqslant \mathbb{E}_{X_1 \cdots X_T} \mathbb{E}_{\sigma_1, \cdots, \sigma_T}\left[\sum_{t \in [T]} \tilde{f}_t(\theta_t) - \inf_{\theta \in \Theta} \sum_{t \in [T]} \tilde{f}_t(\theta)\right].$$

One recognises (up to a factor $T$) the left-hand side of Proposition 4 on the loss sequence $(\tilde{f}_t)_{t \in [T]}$. Furthermore, we can use this proposition as, given our choice of $\mathcal{R}$, for any $t \in [T]$, $\tilde{f}_t$ is $L-$Lipschitz with $L = 5D_{\mathcal{Z}}G(4\alpha)^{-1}$ almost surely by Lemma 1 (with $\chi = 0$). We then have:

$$\mathbb{E}_{\sigma_1, \cdots, \sigma_T}\left[\frac{1}{T} \sum_{t \in [T]} \tilde{f}_t(\theta_t) - \inf_{\theta \in \Theta} \frac{1}{T} \sum_{t \in [T]} \tilde{f}_t(\theta)\right] \leqslant \mathcal{O}\left(\eta m^2 D \frac{1}{\alpha^2} + \frac{mD}{\eta T} + \xi\right). \tag{54}$$

Dividing Equation (53) by $T$, and plugging Equation (54) gives: for all $\theta \in \Theta$:

$$
\begin{aligned}
T^{-1}\mathbb{E}[\mathrm{R}_T^s(\theta)] &\leqslant \mathcal{O}\Big(\eta m^2 D\frac{1}{\alpha^2} + \frac{mD}{\eta T} + \xi\Big) \\
&\quad + 2D_{\mathcal{Z}}\alpha\Big(1 + 2\ln(d)C_0\Big(\ln\Big(\frac{2}{\alpha}\Big) + (1-\beta)\ln^2(\alpha\ln(d))\Big)\Big), \\
&= \tilde{\mathcal{O}}\Big(\eta m^2 D\frac{1}{\alpha^2} + \frac{mD}{\eta T} + \xi + \alpha\ln(d)\Big).
\end{aligned}
$$

Finally remark that, by the definition of the conditional expectation (thus of $F_t$), and because $\theta_t$ is $\mathcal{F}_{t-1}$-measurable, we have for any $\theta$:

$$
\begin{aligned}
\mathbb{E}[\mathrm{R}_T^s(\theta)] &= \mathbb{E}\Big[\sum_{t=1}^T f_t(\theta_t) - f_t(\theta)\Big] \\
&= \mathbb{E}\Big[\sum_{t=1}^T f_t(\theta_t)\Big] - \mathbb{E}[f_t(\theta)]
\end{aligned}
$$

Then taking the infimum over $\theta$ yields:

$$
T^{-1}\mathbb{E}[\mathfrak{R}_T^s] \leqslant \tilde{\mathcal{O}}\Big(\eta m^2 D\frac{1}{\alpha^2} + \frac{mD}{\eta T} + \xi + \alpha\ln(d)\Big).
$$

This concludes the proof. The second equation consists in simply plugging the proposed value of $\eta, \alpha$ in this bound. $\qquad\square$

## G.5 Proof of Proposition 3

**Proposition 3.** *Assume H1, H2, and having access to an $\xi$-approximate optimization oracle $\mathbf{O}_\xi$ adapted to $\big\{\tilde{f}_i\big\}_{t\in[T]}$. Fix $\mathcal{W} = \mathcal{W}_0$, $\mathcal{R} = \mathcal{R}_0$. Let $\{\theta_t\}_{t\in[T]}$ be the output of `DF-OGD` (Algorithm 2) instantiated with the non-increasing sequence $(\eta_t)_{t\in[T]}$ and regularization coefficients $(\alpha_t)_{t\in[T]}$. Then:*

$$
\begin{aligned}
T^{-1}\mathbb{E}\big[\mathfrak{R}_T^d\big] &\leqslant \frac{D_\Theta^2 + D_\Theta\mathbb{E}[P_T]}{2T\eta_T} + \sum_{t\in[T]}\frac{25D_{\mathcal{Z}}G\eta_t}{32T\alpha_t^2} \\
&\quad + 2D_{\mathcal{Z}}\sum_{t\in[T]}\alpha_t[1 + 2\ln(d)C_0\{\ln(2\alpha_t^{-1}) + (1-\beta)\ln^2(\alpha_t\ln d)\}] + \xi\,,
\end{aligned}
$$

*where $P_T = \sum_{t=1}^{T-1}\|\vartheta_{t+1} - \vartheta_t\|$.*

*Proof.* In this proof, for the sake of conciseness, we denote by $\mathbb{E}_t$ the conditional expectation $\mathbb{E}[\cdot \mid \mathcal{H}_{t-1}]$.

Observe that the dynamic regret can be decomposed as follows:

$$
\mathfrak{R}_T^d = \sum_{t\in[T]}\mathbb{E}_t[\langle\bar{g}_t(X_t), w_t^\star(\theta_t) - \tilde{w}_t(\theta_t)\rangle] + \sum_{t\in[T]}\mathbb{E}_t[\langle\bar{g}_t(X_t), \tilde{w}_t(\theta_t) - \tilde{w}_t(\vartheta_t)\rangle] \tag{55}
$$

$$
+ \sum_{t\in[T]}\mathbb{E}_t[\langle\bar{g}_t(X_t), \tilde{w}_t(\vartheta_t)\rangle] - \inf_{\theta\in\Theta}\mathbb{E}_t[\langle\bar{g}_t(X_t), \tilde{w}_t(\theta)\rangle] \tag{56}
$$

$$
+ \sum_{t\in[T]}\Big[\inf_{\theta\in\Theta}\mathbb{E}_t[\langle\bar{g}_t(X_t), \tilde{w}_t(\theta)\rangle] - \inf_{\theta\in\Theta}\mathbb{E}_t[\langle\bar{g}_t(X_t), w_t^\star(\theta)\rangle]\Big] \tag{57}
$$

First, we remark that for any $t$, given $\vartheta_t = \mathbf{O}(\tilde{f}_t)$, we control (56) by Jensen:

$$\mathbb{E}_t[\langle \bar{g}_t(X_t), \tilde{w}_t(\vartheta_t)\rangle] - \inf_{\theta \in \Theta} \mathbb{E}_t[\langle \bar{g}_t(X_t), \tilde{w}_t(\theta)\rangle] \leqslant \mathbb{E}_t\left[\tilde{f}_t(\vartheta_t) - \inf_{\theta \in \Theta} \tilde{f}_t(\theta)\right]$$
$$\leqslant \xi.$$

Then, taking the $\sup$ over $\Theta$ for each summand of the first sum of (55) (which is valid as $\theta_t$ is $\mathcal{F}_{t-1}$-measurable), defining by $\mathrm{Reg}_T := \sum_{t\in[T]} \mathbb{E}_t[\langle \bar{g}_t(X_t), \tilde{w}_t(\theta_t) - \tilde{w}_t(\vartheta_t)\rangle]$, and noticing for (57) that $\inf_{\theta \in \Theta} \mathbb{E}_t[f_t(\theta)] - \inf_{\theta \in \Theta} \mathbb{E}_t[\tilde{f}_t(\theta)] \leqslant \sup_{\theta \in \Theta} \mathbb{E}_t[f_t(\theta) - \tilde{f}_t(\theta)]$ leads to:

$$\mathfrak{R}_T^d \leqslant \mathrm{Reg}_T + 2 \sum_{t\in[T]} \sup_{\theta \in \Theta} \mathbb{E}_t[\langle \bar{g}_t(X_t), w_t^\star(\theta) - \tilde{w}_t(\theta)\rangle] + \xi T$$

Using the fact that $\|\cdot\|_\infty$ is dual to $\|\cdot\|_1$:

$$\mathfrak{R}_T^d \leqslant \mathrm{Reg}_T + 2 \sum_{t\in[T]} \sup_{\theta \in \Theta} \mathbb{E}_t[\|\bar{g}_t(X_t)\|_\infty \|w_t^\star(\theta) - \tilde{w}_t(\theta)\|_1] + \xi T$$

Since for any $t \in [T]$, $\|\bar{g}_t(X_t)\|_\infty \leqslant \|\bar{g}_t(X_t)\|_2 \leqslant D_{\mathcal{Z}}$ almost surely by **H**1-(ii),

$$\mathfrak{R}_T^d \leqslant \mathrm{Reg}_T + 2D_{\mathcal{Z}} \sum_{t\in[T]} \sup_{\theta \in \Theta} \mathbb{E}_t[\|w_t^\star(\theta) - \tilde{w}_t(\theta)\|_1] + \xi T \qquad (58)$$

First, notice that, by Lemma 2, for all $t$:

$$\sup_{\theta \in \Theta} \mathbb{E}_t[\|w_t^\star(\theta) - \tilde{w}_t(\theta)\|_1] \leqslant \alpha_t\left(1 + 2\ln(d)C_0\left(\ln\left(\frac{2}{\alpha_t}\right) + (1-\beta)\ln^2(\alpha_t \ln(d))\right)\right). \qquad (59)$$

Second, taking the expectation (denoted by $\mathbb{E}$) over $(u_t)_{t\in[T]}$ and $X_1, \cdots X_T$ on both sides gives:

$$\mathbb{E}[\mathfrak{R}_T^d] \leqslant \mathbb{E}[\mathrm{Reg}_T]$$
$$+ 2D_{\mathcal{Z}} \sum_{t\in[T]} \alpha_t\left(1 + 2\ln(d)C_0\left(\ln\left(\frac{2}{\alpha_t}\right) + (1-\beta)\ln^2(\alpha_t \ln(d))\right)\right) + \xi T. \qquad (60)$$

To bound the first term in the right-hand side of Equation (60). Remark that, by the definition of conditional expectation:

$$\mathbb{E}[\mathrm{Reg}_T] = \mathbb{E}_{X_1\cdots X_T} \mathbb{E}_{u_1,\cdots,u_T}\left[\sum_{t\in[T]} \tilde{f}_t(\theta_t) - \tilde{f}_t(\vartheta_t)\right].$$

One recognises the definition of $\tilde{\mathfrak{R}}_T^d$ of Lemma 8. Thus, by this lemma, we know that for any $t \in [T]$, because $\tilde{f}_t$ is $K_t$−Lipschitz with $K_t = 5D_{\mathcal{Z}}G(4\alpha_t)^{-1}$ almost surely by Lemma 1, we have:

$$\mathbb{E}_{u_1,\cdots,u_T}\left[\sum_{t\in[T]} \tilde{f}_t(\theta_t) - \tilde{f}_t(\vartheta_t)\right] \leqslant \frac{1}{2\eta_T}(D_\Theta^2 + D_\Theta P_T) + \sum_{t\in[T]} \frac{25D_{\mathcal{Z}}^2 G^2 \eta_t}{32\alpha_t^2}. \qquad (61)$$

Plugging Equation (61) into Equation (60) and then dividing by $T > 0$ on both sides gives the desired result. $\qquad \square$

## G.6 PROOFS OF APPENDIX F

*Proof of Proposition 5.* The regret simply decomposes as follows:

$$
\begin{aligned}
\mathfrak{R}_T^d &= \sum_{t\in[T]} \mathbb{E}_t[\langle \bar{g}_t(X_t), \underline{\mathbf{w}}_t(\theta_t)\rangle] - \sum_{t\in[T]} \inf_{\theta\in\Theta} \mathbb{E}_t[\langle \bar{g}_t(X_t), w_t^\star(\theta)\rangle] \\
&= \sum_{t\in[T]} \mathbb{E}_t[\langle \bar{g}_t(X_t), \underline{\mathbf{w}}_t(\theta_t) - w_t^\star(\theta_t)\rangle] \\
&\quad + \sum_{t\in[T]} \mathbb{E}_t[\langle \bar{g}_t(X_t), w_t^\star(\theta_t)\rangle] - \sum_{t\in[T]} \inf_{\theta\in\Theta} \mathbb{E}_t[\langle \bar{g}_t(X_t), w_t^\star(\theta)\rangle] \\
&\leqslant \kappa + \sum_{t\in[T]} \mathbb{E}_t[\langle \bar{g}_t(X_t), w_t^\star(\theta_t)\rangle] - \sum_{t\in[T]} \inf_{\theta\in\Theta} \mathbb{E}_t[\langle \bar{g}_t(X_t), w_t^\star(\theta)\rangle] \,,
\end{aligned}
$$

and the proof continues as in Theorem 2. □

