# OpenReview forum: "Online Decision-Focused Learning"
_ICLR.cc/2026/Conference — ICLR 2026 Poster_

### Official Review · Reviewer_riB6 · 2025-10-25

**Soundness:** 3
**Presentation:** 2
**Contribution:** 3
**Rating:** 4
**Confidence:** 2

**Summary:**

This paper studies the predict-then-optimize framework in the online setting. To handle the non-stationarity of online data streams, the authors regularize the objective and present two new algorithms, based on FTPL and OGD, to handle the non-convexity of the overall objective function. The authors prove both static and dynamic regret bounds, complementing their theoretical results with strong empirical evidence.

**Strengths:**

- The paper is well-written and easy to follow
- The problem is well-motivated and of general interest to the ML community
- In my opinion, the contributions by the authors are substantial as they provide both strong theoretical and empirical results

**Weaknesses:**

My biggest gripe with this paper is its lack of clarity in the problem setup. Below, I summarize a few points of confusion for me.

-  In line 136, the authors write $\bar{g}_t(X_t) = E[Z_t|X_t]$ for some hidden state $Z_t \n R^d$. What does this mean? Does $X_t$ index a hidden distribution $D_t$ and $\bar{g}_t(X_t)$ is the expected value of this distribution?

- In line 176, $F_t$ is defined in terms of an expectation. What is the expectation taken with respect to? Just the learner's randomness?

- In Equation 4, why is there an expectation over the player's actions in dynamic regret but not static regret?

- In Line 196, what is the probability taken with respect to? Line 164-165 claims that you place no assumption on $(X_t \bar{g}_t(X_t)), so I'm not sure where the randomness is coming from...

-  Likewise, why is there an expectation on the data in Theorem 1 if, according to Lines 164-165 you place no assumptions on the data-generating process?

Overall, I think the authors need to clarify how the adversary is selecting the data stream. This lack of clarity is the main reason for my rating. I am happy to increase my score, given that the authors address this.

**Questions:**

See Weaknesses. Also, can the authors comment on practical settings where having access to an offline optimization oracle is reasonable?

---

> ### Author Response · Authors · 2025-11-20
>
> We thank the reviewer for their positive feedback on the motivation and contribution of our work.
>
> We also understand their concerns regarding the clarity of how the adversary selects that data stream. In a nutshell, $(X_t, \bar{g}_t (X_t))$ IS a random process. At each round, nature picks a distribution for the random variable $X_t$ as well as a fixed (non-random) function $g_t:\mathcal{X}\rightarrow\mathcal{Z}$, adversarially and adaptatively. This corresponds to the stochastic adversary framework [1]. We suspect the confusion comes from line 131, where we wrote *"$X_t\in\mathcal{X}$ are covariates (possibly random)"*, while they are clearly random. We thank the reviewer for asking for this clarification. We detailed it carefully in lines 131-134 and 140 of the revised version.
>
> Accordingly, the conditional expectations appearing in our regret definitions and assumptions are defined with respect to both the randomness of the learner’s algorithm AND the past of the data process $(X_\tau)_ {\tau<t}$. We gathered all this randomness in the notion of history $(\mathcal{H} _ t) _ {t\geq 0}$  defined in line 150-152.
>
> > “In line 136, the authors write…”
>
> Indeed, at each round $X_t$ is a random variable whose distribution may evolve adversarially. The hidden state we are referring to is
> $$
> Z_t = g_t(X_t) + \varepsilon_t,
> $$
> for some centered noise $\varepsilon_t$, so that $\mathbb{E}[Z_t \mid X_t] = g_t(X_t)$. This hidden state stems from the classic linear decision-focused setting, see for instance [2].
>
> > “In line 176, F_t is defined in terms of an expectation…”
>
> The $F_t$ appearing in our dynamic regret is an expectation conditioned on $\mathcal{H}_{t-1}$, which we discussed above.
>
> > “In Equation 4, why is there an expectation over the player's actions in dynamic regret but not static regret?
>
> In our dynamic regret, the action sequence is compared to a sequence of oracle decisions that minimize the instantaneous loss. Without taking the conditional expectation over $\mathcal{H}_{t-1}$, each comparator could overfit to the realized $X_t$, creating an unrealistically strong benchmark. Using the conditional expectation regularizes these comparators and makes the dynamic regret meaningful; we clarify this in lines 175–180 of the revised version.
>
> We also note that we could equivalently use $F_t$ in the static regret, since $f_t$ and $F_t$ coincide once averaged over the process and the algorithm’s randomness, which is the case in Theorem 1. We kept $f_t$ for clarity, but are happy to adjust this if the reviewer prefers.
>
> > “In Line 196, what is the probability taken with respect to? Line 164-165 claims that you place no assumption on $(X_t \bar{g}_t(X_t)), so I'm not sure where the randomness is coming from...”
>
> In line 196, the probability is taken with respect to the distribution of $X_t$ (and thus $I_t$) conditioned on $\mathcal{H}_{t-1}$. We agree that lines 164--165 may not have been sufficiently clear. Our intent was simply to state that no stationarity, independence, or identical-distribution assumptions are required: the sequence $(X_t, \bar{g}_t(X_t))$ may be arbitrary as long as H2 holds. This does not imply the process is non-random. We clarified this point in lines 165-167 of the revised manuscript.
>
> > “Likewise, why is there an expectation on the data in Theorem 1 if, according to Lines 164-165 you place no assumptions on the data-generating process?”
>
> We believe that this point is similar to the one above. $(X_t, \bar{g}_t, (X_t))$ are indeed random variables, and the expectation is taken over the process randomness alongside the randomness of the algorithm yielding $(\theta_t){t\geq 0}$..
>
> >> “ Also, can the authors comment on practical settings where having access to an offline optimization oracle is reasonable?”
>
> Invoking the oracle via an SGD procedure (for which a theoretical justification is provided in Appendix B) is not particularly expensive from a computational point of view. In our experiment, we use warm starts so that, as long as the loss landscape does not change dramatically from one round to the next, the descent is initialized near a local minimizer. This yields very good performance in a constant number of iterations at each round.
>
> We would also like to highlight that, in most practical industrial settings, solving the downstream optimization problem is typically far more computationally demanding than updating the prediction model. It is therefore reasonable to assume that the oracle call represents only a small fraction of the overall computational cost. We added this remark in the revised version, lines 814-817.
>
> We hope that the above answers address the reviewer’s concerns and clarify how the data process is chosen. We remain available for any additional comment.
>
> [1] A. Rakhlin (2011) Online learning: Stochastic, constrained, and smoothed adversaries.
> [2] Sadana, U (2023). A survey of contextual optimization methods for decision making under uncertainty.

---

> > ### Comment · Reviewer_riB6 · 2025-11-26
> >
> > I thank the authors for their reply and their changes to the manuscript. All my concerns have been addressed, and I have increased my score accordingly.

---

### Official Review · Reviewer_FEZs · 2025-10-27

**Soundness:** 3
**Presentation:** 3
**Contribution:** 2
**Rating:** 4
**Confidence:** 3

**Summary:**

The paper studies an online version of decision-focused learning, where the learner is evaluated not by the accuracy of the predictions but by the quality of the decisions made. Unlike in the batch case, in the online version, the problem may evolve over time in a fully non-stationary way. As the problem is non-differentiable and non-convex, the authors leverage a regularized version of the bi-level optimization problem and rely on near-optimal oracles to derive regret guarantees by instantiating follow-the-leader and online gradient descent algorithmic schemes. Finally a synthetic experiment illustrates the effectiveness of the proposed algorithms in minimizing the average cost.

**Strengths:**

* The paper extends the decision-focused framework to online learning, whereas most of the existing literature on the topic is limited to the batch case, where iid samples and a fixed estimation/optimization problem is available.
* The authors provide a complete analysis of the problem leveraging state-of-the-art technical algorithms and tools from online learning literature.

**Weaknesses:**

* The technical novelty of the paper is limited. While the authors stress the challenges posed by the online decision-focused learning setting (non-differentiable and non-convex functions), once formulated as in eq.(3) the problem is amenable to any "standard" online learning treatment. Indeed most of the results in the paper are obtained by carefully instantiating known assumptions, algorithms, and theoretical results, such as assumptions H1/H2, FTL and OGD, regularization and approximate oracle for non-convex objectives. As such, the results in the paper should be mostly assessed based on the interest of the setting.
* While I'm not very familiar with the decision-focused learning paradigm, I can see a strong resemblance with other decision-making settings in machine learning, where accuracy in prediction problems do not directly translate into performance wrt to a target objective function. These includes reinforcement learning (e.g., accurately estimating an MDP does not translate into computing an optimal policy for a given reward) or bandit (e.g., accurately estimating the mean of each arm does not translate into accurately returning a high-reward arm). While the decision-focused learning paradigm may have emerged in a different literature, it would be helpful to have a more extensive justification of how the proposed online version differs from adversarial bandit/RL settings and provide more practical examples supporting its relevance.

**Questions:**

* Please refer to weaknesses.
* Please clarify if any technical contribution in the paper is novel or it required specific treatment due to the nature of the problem.
* In Fig.1 the average cost of PFL, SPO, and FTPL see to converge towards the same value. Is this the case? If the overall process is stationary and ergodic, maybe we should indeed expect all methods to coverage to the same performance. If not, what is the explanation behind the difference in performance?

---

> ### Author Response · Authors · 2025-11-20
>
> We thank the reviewer for their feedback, and reply to their remarks and questions below.
>
> > “”The technical novelty of the paper is limited [...] Please clarify if any technical contribution in the paper is novel or if it required specific treatment due to the nature of the problem [...]”
>
> We humbly disagree with the reviewer about our analysis being standard. We reply to their concerns below.
>
> -> *“once formulated as in eq.(3) the problem is amenable to any "standard" online learning treatment”*
> As explained in lines 215--290, problem (3) is both non-differentiable and non-convex. Contrary to the reviewer’s suggestion, this places us in a highly non-standard setting where standard first-order online methods fail due to the absence of gradients, and even after smoothing, non-convexity prevents a classical analysis based on distance to a global minimizer. Our contribution is precisely to overcome these difficulties by reducing the problem to a surrogate that can be handled with known tools—a step that is technically delicate and requires substantial work.
>
> -> *“most of the results in the paper are obtained by carefully instantiating known assumptions, algorithms, and theoretical results.”*
>
> Here again, we respectfully disagree that our work is merely a sequence of known results. Handling the non-convex, non-differentiable problem (3) required several new ideas:
>
> (i) For general polytopes, we introduce a log-barrier regularization and use the implicit function theorem to recover gradients. In Lemmas 1 and 5 we show, via a new analysis, that due to the bilevel structure, this strongly convex regularizer yields only Lipschitz continuity. This subtle effect drives the trade-off in the parameters $(\alpha_t)$ and leads to our $\mathcal{O}(T^{-1/4})$ rate rather than the usual $\mathcal{O}(T^{-1/2})$.
>
> (ii) We make an original use of the margin assumption H2, together with log-barrier properties, to control the expected gap between the regularized and true minimizers (Lemmas 2 and 6), which is key to ensuring that the surrogate problem faithfully approximates the original one.
>
> (iii) We propose a new OGD variant with a dynamic-regret analysis for non-convex objectives (Lemma 8). By evaluating the gradient at a random point on the segment to the oracle solution and combining this with offline approximate oracles, we overcome non-convexity. To our knowledge, this is the first gradient-based method that blends these ingredients and achieves meaningful dynamic regret guarantees.
>
> > “While I'm not very familiar with the decision-focused learning paradigm, I can see a strong resemblance with other decision-making settings in machine learning [...]”
>
> We thank the reviewer for this very interesting point. We agree that the gap between accurate prediction and good decisions is not unique to DFL, it also appears in bandits and RL. The value of (online) DFL typically emerges in problems where the covariate space $X_t$ is much higher dimensional than the state space $Z_t$. This is for instance common in power-system applications such as energy storage [1] or optimal dispatch [2], where many high-dimensional covariates (weather, unit conditions, etc.) ultimately inform a low-dimensional state (e.g., a 24-hour price vector). In such settings, contextual bandits would face an exponential dependence on the covariate dimension. On the contrary, DFL learns a model mapping covariates to states, so the complexity depends only on the model’s parameter dimension, which can be far smaller.
>
> > “In Fig.1 the average cost of PFL, SPO, and FTPL see to converge towards the same value. Is this the case?”
>
> In our experiments, the average costs of PFL, SPO, and FTPL do not converge to the same value. Figure 1 reports 95\% Gaussian confidence intervals (over 10 runs), and these intervals do not overlap, allowing us to reject convergence to a common cost. Although the data-generating process is stationary, the learning procedures differ, which explains the persistent performance gaps:
>
> – PFL vs. DFL: PFL optimizes prediction accuracy, whereas DFL directly minimizes decision regret, leading to lower final costs.
>
> – SPO vs. our methods: Online SPO optimizes the $\mathrm{SPO}^+$ surrogate of the latest cost, and this myopic update can drift even under stationarity. Our optimistic, oracle-based updates better track the objective.
>
> – DF-FTPL vs.\ DF-OGD: FTPL queries the oracle on cumulative loss, providing extra stability and a slight advantage over OGD, which relies only on the most recent cost.
>
> We hope to have addressed the reviewer’s concern regarding the technical novelty of our approach and its originality as compared to other learning procedures. We are happy to answer any additional questions they might have.
>
> [1] Sang, L. (2022). Electricity price prediction for energy storage system arbitrage: A decision-focused approach.
>
> [2] Wahdany, D. (2023). More than accuracy: end-to-end wind power forecasting that optimises the energy system

---

### Official Review · Reviewer_X6pL · 2025-11-01

**Soundness:** 2
**Presentation:** 3
**Contribution:** 2
**Rating:** 6
**Confidence:** 3

**Summary:**

This paper studies decision-focused online learning and utilize regularization for differentiability and perturbation for non-convexity. They succeed in generating static and dynamic regret results with respect to loss predictor parameters.

**Strengths:**

Originality:
The originality arises from a proper problem formulation for the regret analysis of decision-focused online optimization, incorporating static and dynamic regret analysis into the said problem and removing limitations of non-differentiability and non-convexity.

Quality:
The submission seems technically correct, experimentally rigorous and reproducible (except minor caveats in the algorithms).

Clarity:
The submission is mostly clear.

Significance:
The submission presents theoretical (and possibly algorithmic) novel findings to achieve the static and dynamic regret results for online decision-focused learning.

**Weaknesses:**

I did not notice any substantial weakness, so I am leaning towards acceptance.

One thing to note is that the exact challenge in achieving the said results could be emphasized. Possibly, after proper formulation, everything follows from the existing regret analysis techniques.

**Questions:**

Questions:

Page 2 Line 93: how does the linear structure cause zero or undefined gradient?

Page 4 Line 195: should it be $i\in[K]$?

Page 4 Line 204: why does $\epsilon=0$ imply anything? Doesn't the inequality hold for all $\epsilon$?

Page 6 Algorithm 1 Line 3: shouldn't the objective for $w$ be regularized?


Suggestions:

Page 2 Line 85: correct grammar.

Page 3 Line 117: there seems to be a typo between $g$ and $\phi$.

Page 5 Line 275: $\Theta$ was the parameter space in (8). Please correct.

Page 7 Line 352: check grammar.

---

> ### Author Response · Authors · 2025-11-20
>
> We deeply thank the reviewer for their positive feedback and their remarks, to which we answer below.
>
> > “One thing to note is that the exact challenge in achieving the said results could be emphasized. Possibly, after proper formulation, everything follows from the existing regret analysis techniques.”
>
> From our point of view, the main challenge of online DFL lies in first formulating an online, non-convex, and non-differentiable bilevel problem in a way that known tools can be meaningfully applied. To our knowledge, this combination of difficulties has not been addressed before, and bridging the gap between the bilevel structure and online-regret methods is itself a nontrivial step.
>
> We also note that our analysis does not follow automatically from existing results. In particular, we introduce a tailored OGD-style update (studied in Lemma 8) that evaluates gradients at a random point along the segment between the last iterate and the oracle output, which enables us to control non-convexity in ways not covered by standard analyses. In addition, our treatment of log-barrier and entropy regularization shows that, unlike in single-level problems where these regularizers induce strong convexity, in the bilevel setting they provide only Lipschitz continuity. This distinction creates a delicate trade-off in the choice of the regularization schedule and leads to our $\mathcal{O}(T^{-1/4})$ rate instead of the $\mathcal{O}(T^{-1/2})$ rate typical in simpler settings.
>
> > “Page 2 Line 93: how does the linear structure cause zero or undefined gradient?”
>
> Since $w^\star_t (\theta)$ minimizes a * linear * problem (namely, \min_{w\in\mathcal{W}}\langle g(\theta, X_t), w\rangle), it lies on a vertex of the feasible polytope. Intuitively, a small change in $\theta$ can change the direction of $g(\theta, X_t)$ and make $w^\star _t (\theta)$ jump to another vertex abruptly. All in all, $\nabla f_t (\theta)$ is either zero (if $w^\star _t (\theta)$ remains on the same vertex in spite of the infinitesimal change $d\theta$) or undefined (if $w^\star _t (\theta)$ jumps to another vertex because of $d\theta$.)
>
> > “Page 4 Line 195: should it be $i\in[K]$?”
>
> This is indeed a typo that we corrected in the revised version. We thank the reviewer for spotting it.
>
> > “Page 4 Line 204: why does $\varepsilon=0$  imply anything? Doesn't the inequality hold for all $\varepsilon$?”
>
> For $\varepsilon=0$, the bound on the probability is equal to 1, and thus trivial. The assumption is only meaningful for $\varepsilon>0$. At a high level, the assumption avoids hard cases where the suboptimality gap is very small with significant probability.
>
> > “Page 6 Algorithm 1 Line 3: shouldn't the objective for $w$ be regularized?”
>
> We thank the reviewer for this interesting remark. Line 3 displays the *actual* action taken by the decision-maker, which need not to be regularized: this is the best possible decision given the predicted cost $g(\theta_t, X_t)$. In fact, we only need to regularize it when updating $\theta_t$ to $\theta_{t+1}$, so $\tilde{f}_t$ admits a gradient. We made this fact clearer in line 317-319 of the revised version.
>
> > “Suggestions: [...]
> We thank the reviewer deeply for these suggestions, which we included in the revised version.
>
> We hope to have replied to most of the reviewer’s concerns, and remain available for further clarifications.

---

### Official Review · Reviewer_JCMo · 2025-11-01

**Soundness:** 4
**Presentation:** 4
**Contribution:** 4
**Rating:** 10
**Confidence:** 4

**Summary:**

This paper tackles the problem of training predictive models that are used not downstream decision-making tasks. The authors frame this as a bi-level online optimization problem. They overcome challenges of non-differentiability and non-convexity by using regularization and perturbation, and develop variants of follow the perturbed leader and online gradient descent that get sublinear static and dynamic regret.

**Strengths:**

This is an excellent paper!

* The paper is very well written and includes intuitive explanations along with precise mathematical statements.

* The paper tackles a fundamental problem: online decision-focused learning, where the goal is to train predictive models not just for prediction but for using those predictions in a downstream decision-making task. While prior work looked at this problem in the offline setting, this paper studies this problem in the online setting, provides provable regret guarantees - static and dynamic - and shows strong experimental results, where the proposed algorithms have a worse prediction accuracy than baselines but superior downstream performance.

* The formulated problem is challenging: bi-level optimization, non-differentiable, and non-convex. The authors overcome these challenges using a combination of smoothing/regularization, perturbation and offline optimization oracles - all somewhat standard tools. Then, they develop variants of standard online learning algorithms, follow the perturbed leader and online gradient descent, for the online decision-focused setting. However, the overall problem formulation and solution methodology is very elegant and this is a plus in my opinion, especially when combined with the theoretical and empirical results and the importance of the problem.

**Weaknesses:**

Please see the questions section some weaknesses/questions.

**Questions:**

* The oracle assumption is understandable but it seems quite expensive to invoke the oracle in each round, especially for large non-convex problems. I understand that there are a lot of analyses on favorable loss landscapes for neural networks and how gradient descent-based methods can find "good optima". However, running such methods to convergence in each round seems quite slow in practice. Did you measure this in your experiments / are your experiments large-scale enough for this to be a problem? Do you have thoughts on how you could use lazy updating to only invoke the oracle once every few rounds while maintaining or suffering from a tolerable degradation in the regret bound?

* Is my understanding correct that the choice of the regularizer does not affect the final regret bound as long as the hyperparameter $\alpha_t$ is chosen correctly? This hyperparameter is chosen a function of $T, m$ and $n$. I understand that the time horizon is sometimes unknown. Are there cases where $m$ and $n$ might be unknown? In case of mis-specified $T, m$ or $n$, how does the regret bound degrade with the degree of mis-specification?

* The paper assumes that the set $\mathcal{W}$ is a convex polytope. How restrictive is the polytope assumption? If we had general convex sets, how would that impact your analyses and where would the problems arise?

* The (average) regret bound of $ T^{-\frac14} $ is quite slow compared to the usual $ T^{-\frac12} $ regret bounds. I understand that the problem in this paper is harder (bi-level optimization, non-differentiable, non-convex). Do you have thoughts on whether this is tight and a lower bound? If not, what do you think could lead to improved regret bounds? For example, using an alternative to perturbation, an alternative to offline optimization oracles, etc.?

---

> ### Author Response · Authors · 2025-11-20
>
> We deeply thank the reviewer for their extremely positive feedback, as well as their very interesting questions. We answer below to these questions.
>
> > “The oracle assumption is understandable but [...]?”
>
> The computational cost of calling our oracle is of prime importance to understand the per-iteration runtime of our procedure, so we are happy to clarify this point.
> In our experiments, DF-OGD and DF-FTPL are indeed slightly slower than PFL or online SPO (see the table below for average runtimes (in seconds) over 5 runs with $T=500$), but we consider that this overhead is acceptable given their performance gains. We also note that in typical OR settings targeted by decision-focused learning, downstream optimization is far more computationally intensive than updating the prediction model, so oracle calls usually account for only a small share of total runtime.
>
> At each round, only $k=10$ gradient steps are required to compute $\vartheta_t$ while still outperforming PFL and SPO. This efficiency comes from warm starts: as long as the loss landscape changes slowly across rounds, each descent begins near a good local minimizer, allowing the oracle to converge in a small, fixed number of steps.
>
> We also did not observe any meaningful runtime increase as the number of items grew (see the table). Importantly, the relative performance gains over PFL and SPO persist at larger scales. An additional plot for $80$ items appear in Appendix E of the revised version.
>
> | N items | PFL | DF-OGD | DF-FTPL | SPO  |
> |--------|-------|----------|-----------|-------|
> | 5 | 0.043 | 0.600 | 1.136 | 0.052 |
> | 10 | 0.052 | 0.683 | 1.241 | 0.055 |
> | 20 | 0.063 | 0.907 | 1.571 | 0.058 |
> | 40 | 0.047 | 0.606 | 1.460 | 0.057 |
> | 80 | 0.044 | 0.596 | 1.278 | 0.061 |
>
>
> > “Is my understanding correct that the choice of the regularizer [...]?”
>
> The choice of regularizer in fact affects the final regret bound. In the main text we analyze the general polytope case with log-barriers, while Appendix C treats the simplex with negative entropy, yielding a tighter bound (replacing the dependence on the number of faces $n$ with a $\ln(d)$ term).
> The reviewer righfully points out that the dependence of $\alpha$ on $m$ and $n$ is important. Although these quantities are typically known in practice, ignoring them prevents use of the optimal rate. In such cases, we suspect that good performance could still be achieved by tuning the two parameters $\alpha$ and $\eta$ via a grid search. If $T$ were unknown, the doubling trick could be applied with slightly worse constants. We note that this last remark about $T$ applies only to DF-FTPL and static regret, since DF-OGD adapts its parameters online.
>
> > “The paper assumes that the set $\mathcal{W}$ is a convex polytope [...]?”
>
> We thank the reviewer for this deep and insightful question. We address it by distinguishing two situations:
>
> (i) *$\mathcal{W}$ has a smooth boundary.*
> In this case, $w_t^\star(\theta)$ would vary smoothly with $\theta$ rather than jumping between vertices. It would be differentiable, and differentiating the KKT conditions associated to the problem would recover $\nabla w_t^\star(\theta)$ directly. This setting is therefore simpler than the polytope case, as no regularization would be required before applying a descent step.
>
> (ii) *$\mathcal{W}$ has a non-smooth boundary (e.g., a ball intersected with a half-space).*
> Our current analysis would not extend immediately, since H2 relies on minimizers lying among the polytope’s extremal points. Although adapting H2 might be possible, one alternative is to rewrite constraints through indicator functions and apply a Moreau–Yosida regularization to obtain a smooth proxy objective. This leads naturally to proximal-gradient-style updates, for which online guarantees already exist [1]. We view this as a promising direction for future work.
>
> We chose to focus on polytopes because they form a very common and well-studied class in OR, covering problems such as shortest path, min-cost flow, and portfolio optimization.
>
> > “The average regret bound of $T^{-1/4}$ is slow compared to $T^{-1/2}$ [...]?”
>
> Actually, we think that a faster rate (e.g., $T^{-1/2}$) might be achievable using traditional online learning methods that avoid the bilevel framework. For example, we could discretize the parameter space and then run experts aggregation on the finite number of parameters bin. Although such a technique might yield a $T^{-1/2}$ under assumptions similar to ours, it would most probably suffer a prohibitive dependency  in the parameter dimension $m$, e.g. with either a regret or computational cost scaling exponentially with $m$ (as the number of bins/experts would scale exponentially with $m$). We added this discussion in lines 486-491 of the revised version.
>
> We sincerely appreciate your thoughtful feedback, and remain available for other discussions.
>
> [1] Dixit, R (2019). Online learning with inexact proximal online gradient descent algorithms.

---

### Author Response · Authors · 2025-12-04
**Summary of discussions**

Dear all,

To facilitate the review of our paper, we provide below a summary of the reviewers’ assessments of our work and an overview of the ensuing discussions.

**Reviewer JCMo**  was particularly positive about our contribution, **giving a score of 10** with confidence 4. They emphasized the importance of the research question, the technical challenges addressed, and the significance of both the theoretical and empirical findings. They raised one question regarding the computational complexity of our method, to which we responded with additional experiments (now included in the revised version) demonstrating that the algorithm scales well with problem dimension. They also asked whether our results could be generalized without knowledge of certain hyperparameters, without assuming that $\mathcal{W}$ is a polytope, and whether our rate is tight. We addressed all three questions and incorporated the corresponding discussions into the revision.

**Reviewer X6pL** also provided positive feedback, noting that our contribution is significant, novel, and technically solid, and **gave a score of 6** with confidence 3. They stated that they “did not notice any substantial weakness.” They suggested emphasizing the technical challenges we overcame and pointed out several typos and minor formulations that could be improved. We addressed these points in the revised version, although we did not have the opportunity to discuss them further with the reviewer.

**Reviewer rib6** found the paper “well-written and easy to follow,” the problem “well-motivated and of general interest to the ML community,” and the contributions “substantial,” given the strong theoretical and empirical results. However, they initially had concerns about the clarity of the adversary’s data-generating process and therefore **initially assigned a score of 4** with confidence 2. After our discussion (and with the additional explanations now included in the revised version) they were satisfied with our clarification and **increased their score from 4 to 6**.

**Reviewer FEZs** expressed concerns that “the technical novelty of the paper is limited” and that the work resembles existing decision-making settings in machine learning, and **gave a score of 4** with confidence 3. In response, we highlighted both the difficulty and novelty of our setting (online, non-differentiable, and non-convex learning) as well as the new algorithmic ideas we introduced to address these challenges. Regarding similarities with frameworks such as bandits or RL, we emphasized the distinctive nature of decision-focused learning, which is designed for problems where the covariate space is typically much higher-dimensional than the state space, making it more efficient than alternative approaches. **The reviewer did not respond to our rebuttal, so we were unable to further engage with their concerns.**

We hope this summary is helpful and remain available for any further clarification.

---

### Note · Authors · 2026-01-26

**Comment:**

The paper has been accepted at ICLR 2026.

**Withdrawal Confirmation:**

I have read and agree with the venue's withdrawal policy on behalf of myself and my co-authors.

---

> ### Note · Program_Chairs · 2026-01-26
>
> We approve the reversion of withdrawn submission.

---

### Meta-Review · Area_Chair_HVoc · 2026-01-07

**Summary:**

This paper addresses the predict-then-optimize problem in an online setting. Since the objective function may be non-convex and non-differentiable, and the data distribution may evolve over time, this setting is particularly challenging. The Authors propose several techniques to tackle these difficulties and derive provable guarantees for the resulting approach.

Two Reviewers are very positive about the paper and raised only minor comments. The other two reviewers expressed more critical concerns, mainly regarding potentially limited novelty and a lack of clarity in the problem setup. The authors’ responses are convincing and adequately address these concerns, making the paper worth publishing.

**Reviewer Concerns:**

All the concerns have been successfully addressed by the rebuttal.

**Reviewer Scores:**

- Reviewer JCMo gave the highest possible score and would likely maintain it.
- Reviewer X6pL would most likely increase their initial score of 6.
- Reviewer rib6 was among the more critical reviewers, but acknowledged the authors’ responses and expressed willingness to increase their score.
- Reviewer FEZs did not respond to the rebuttal, but would likely increase their score.

---

### Decision · Program_Chairs · 2026-01-26

Accept (Poster)